# VideoZoomer: Reinforcement-Learned Temporal Focusing for Long Video Reasoning

**Yang Ding**[1*] **Xin Lai**[2*†] **Yizhen Zhang**[1*] **Wei Li**[3] **Ruihang Chu**[1] **Yujiu Yang**[1‡]
[1]Tsinghua University    [2]The Chinese University of Hong Kong    [3]ByteDance

## Abstract

Multimodal Large Language Models (MLLMs) have achieved remarkable progress in vision-language tasks yet remain limited in long video understanding due to the limited context window. Consequently, prevailing approaches tend to rely on uniform frame sampling or static pre-selection, which might overlook critical evidence and unable to correct its initial selection error during its reasoning process. To overcome these limitations, we propose VideoZoomer, a novel agentic framework that enables MLLMs to dynamically control their visual focus during reasoning. Starting from a coarse low-frame-rate overview, VideoZoomer invokes a temporal zoom tool to obtain high-frame-rate clips at autonomously chosen moments, thereby progressively gathering fine-grained evidence in a multi-turn interactive manner. Accordingly, we adopt a two-stage training strategy: a cold-start supervised fine-tuning phase on a curated dataset of distilled exemplar and reflection trajectories, followed by reinforcement learning to further refine the agentic policy. Extensive experiments demonstrate that our 7B model delivers diverse and complex reasoning patterns, yielding strong performance across a broad set of long video understanding and reasoning benchmarks. These emergent capabilities allow it to consistently surpass existing open-source models and even rival proprietary systems on challenging tasks, while achieving superior efficiency under reduced frame budgets. The code are avaliable at https://github.com/zsgvivo/VideoZoomer.

## 1 Introduction

With a clear task in mind, humans can efficiently navigate long and complex visual streams by dynamically allocating attention, selectively identifying salient events such as decisive actions in a sports match or key explanations in a lengthy lecture, while filtering out redundancy. This goal-directed ability underlies effective and efficient visual reasoning, as widely documented in cognitive science (Kietzmann et al., 2018), remains difficult to achieve in artificial intelligence. Although MLLMs perform strongly on image (Bai et al., 2025; Chen et al., 2024) and short-video tasks (Zhang et al., 2023), they remain constrained in long-video comprehension tasks mainly due to their limited context window (OpenAI, 2024; Reid et al., 2024).

The most common strategy to address this challenge is uniform frame sampling (Zhang et al., 2024b;c), which selects frames at fixed intervals (e.g., two frames per second) to construct a subset that fits within context window. Nevertheless, this strategy is inherently limited, as it assumes all moments are equally important and further risks overlooking short but critical events while allocating context budget to redundant clip segments. To address these limitations, prior work has investigated adaptive frame selection (Yu et al., 2024; Hu et al., 2025a; Tang et al., 2025), where a lightweight selector module, conditioned on the text query, identifies salient frames before reasoning. While improving over uniform sampling, these methods are still inefficient because they are designed to select a fixed number of frames regradless of the problem's complexity. Second, the design remains static and non-interactive. If the initial choice is suboptimal or misses key details,

---

[*]Equal contribution.

[†]Project leader.

[‡]Corresponding authors.

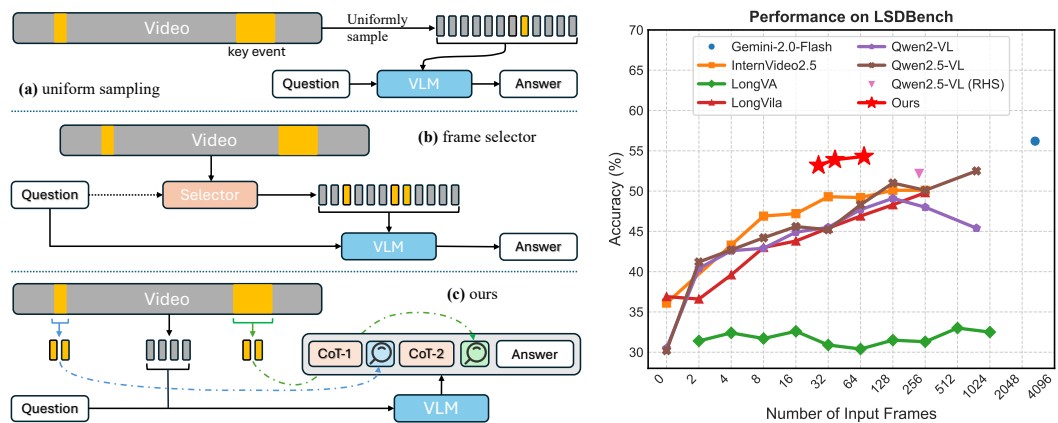

Figure 1: **Left**: Conceptual comparison of three long video reasoning frameworks: (a) uniform sampling, (b) with frame selector, and (c) VideoZoomer (Ours). **Right**: Performance comparison of VideoZoomer against various baseline models under different frame budgets on LSDBench.

the model has no mechanism to correct the error or revisit the video. This fundamentally limits its performance on complex tasks that require iterative evidence gathering.

To overcome the rigidity and inefficiency of prior methods, we propose VideoZoomer, a novel framework that empowers an MLLM to autonomously and dynamically control its visual focus during its reasoning process. As illustrated in Figure 1 (Left), instead of being a passive recipient of pre-selected frames, our model acts as an active agent.

This yields two primary advantages: *(i)* It is highly efficient: the agent begins with a coarse overview of low frame rates, only consuming a significant context budget when it decides to invoke a `<video_zoom>` tool. This on-demand approach ensures that the model's context window is used dynamically and judiciously. *(ii)* It is more performant: by learning a policy to request high-frame-rate clips of specific moments, the agent can correct initial oversights and gather detailed evidence precisely when and where it is needed. This dynamic, iterative evidence-gathering process avoids the critical information loss inherent in static methods and raises the upper bound on reasoning performance. Figure 1 (Right) demonstrates the practical benefit of this design on LSDBench (Qu et al., 2025), a benchmark specifically designed to test a model's ability to find short, critical events in long videos. Our method achieves a better performance-efficiency trade-off, achieving superior accuracy compared to open source baselines while operating on a flexible and smaller frame budget.

Notably, training such an agent faces several challenges, a naive reinforcement learning approach would suffer from an inefficiently large action space and exhibit limited reasoning patterns. To address this, we introduce a two-stage training strategy. First, a cold-start Supervised Fine-Tuning (SFT) phase teaches the model the basics of tool using. Using a tailored dataset of exemplar trajectories, we train the model to understand the task format, master the syntax of tool calls, and develop a baseline reasoning capability. Crucially, to prevent the model from merely imitating a single, monotonous reasoning pattern, we enrich this dataset with reflection data, which exposes our model to more diverse and sophisticated problem-solving strategies. Second, with these foundational skills established, a Reinforcement Learning (RL) phase optimizes the model's tool interaction policy and reasoning capability, transforming it from a simple imitator into an adaptive agent that can generalize its strategy to unseen videos and questions.

We summarize our contributions as follows:

- We propose VideoZoomer, a novel framework that reframes long video understanding as a sequential tool interaction task, enabling an MLLM to dynamically control its visual focus via multi-turn tool interaction.
- We introduce a robust, two-stage training strategy: a cold-start phase using a tailored dataset of exemplar and reflection trajectories, followed by a reinforcement learning phase to optimize an efficient and effective agentic policy.

- We demonstrate through extensive experiments that our model significantly outperforms existing open-source models on a wide range of long video understanding and reasoning benchmarks, in some cases even surpassing leading proprietary models with greater efficiency.

## 2    RELATED WORKS

**Multimodal Reasoning Models.**    The remarkable success in LLMs (Guo et al., 2025; Team et al., 2025; Tan et al., 2025; Jaech et al., 2024; Yang et al., 2025) has demonstrated that reinforcement learning (RL) is a powerful paradigm for enhancing the complex reasoning capabilities. Many works since then have tried to transfer this into multimodal domain. Methods such as MM-Eureka (Meng et al., 2025) and VL-Rethinker (Wang et al., 2025) have successfully adapted RL techniques to improve the vision-language reasoning abilities of MLLMs (Zhang et al., 2025e;b; Luo et al., 2025; Zhang et al., 2025d). More recently, Video-R1 (Feng et al., 2025) further validated the efficacy of this approach specifically within the video domain. Recently some works has tried further extend MLLMs with external tools like image cropping (Zheng et al., 2025; Su et al., 2025; Ma et al., 2026), web search (Wu et al., 2025), segmentation (Liu et al., 2025b). However, most of these methods focus on image tasks and only interact with the environment for single turn, combining RL-driven reasoning and multi-turn tool use strategy for long video understanding is still underexplored.

**Long Video Comprehension.**    Many works have tried to extend the ability of MLLMs in long video comprehension. A stream of research aims to reduce the number of visual tokens that need to be fed into the MLLM (Liu et al., 2025a; Yan et al., 2025) through compression or selection modules.

A second, related approach focuses on selecting a sparse subset of the most salient frames from the entire video. Unlike uniform sampling, these methods aim to identify moments of high importance (Tang et al., 2025; Hu et al., 2025a; Wang et al., 2024b). While effective, the primary limitation of these methods is that the frame selection process is decoupled with its reasoning process, hindering it from learning more complex reasoning patterns. Methods like LongVILA-R1 (Chen et al., 2025) focus on direct context extension by continuing training on long video datasets to handle longer video sequences. Recently, a promising direction has emerged that leverages the powerful zero-shot capabilities of large proprietary models to act as agents. Frameworks like VideoDeepResearch (Yuan et al., 2025) and Deep Video Discovery (Zhang et al., 2025c) use prompting techniques to guide a strong LLM like Deepseek-R1 (Guo et al., 2025) or GPT 4.1 (OpenAI, 2024) to iteratively explore a video with external tools. These training-free methods demonstrate the potential of agentic approach but rely on resource-intensive, closed-source models, making them difficult to optimize, reproduce, or deploy. In contrast, our work focuses on explicitly training a relatively small 7B open-source model to learn an efficient, agentic policy for long video comprehension.

## 3    METHOD

### 3.1    OVERVIEW

To address the challenge of efficient long video understanding with a constrained frame budget, we propose a novel framework, VideoZoomer, which empowers a large multimodal model to actively seek high-temporal-resolution information by invoking an external tool. Rather than relying on fixed or uniform sampling strategies, our model learns to dynamically and adaptively allocate its frame budget during its reasoning process. The core idea is to train an agent that learns an optimal policy for when and where to request high-frame-rate video clips, a process we call "temporal zoom-in", to gather sufficient evidence for answering a given question.

As illustrated in Figure 2, the strategy is summarized as "first glance, then zoom": initially the model only has access to the query prompt $Q$ and a relatively low frame rate version of the video $V_{low}$, uniformly sampled as a default frame rate $f_{low}$, which provides a coarse, computationally inexpensive overview of the entire video. To answer the question accurately, especially when it pertains to fine-grained temporal events or rapid motions, the model may require more detailed visual information. We introduce a <video_zoom> tool, which allows the model to request a specific time segment $[t_{start}, t_{end}]$ from the original video at a higher frame rate, $f_{high}$. Upon

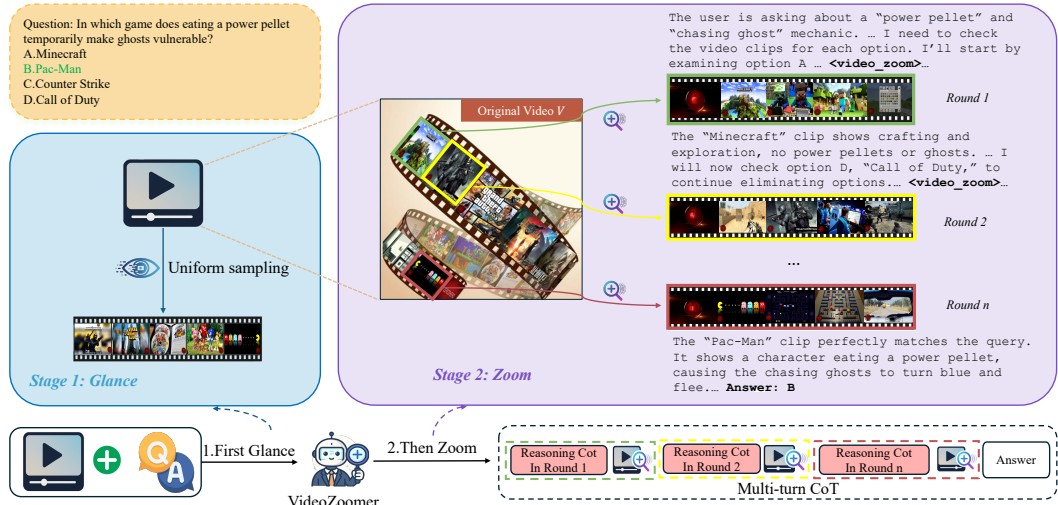

Figure 2: **VideoZoomer framework for long video reasoning.** The process begins with a "Glance" where the model obtains a coarse overview of the video. It then enters an iterative "Zoom" phase, where it can invoke a `<video_zoom>` tool to request high-fps clips and perform multi-turn reasoning. This process continues until the model procudes a final answer or reaches max turn limit.

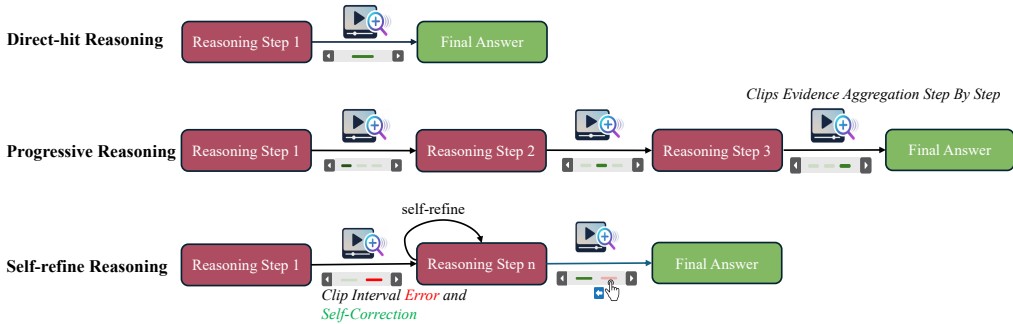

Figure 3: **Diverse reasoning patterns demonstrated by our model.** (a) Direct-hit Reasoning, (b) Progressive Reasoning, and (c) Self-refine Reasoning.

invoking this tool, the environment returns a high-resolution clip $V_{clip} = T(V, t_{start}, t_{end}, f_{high})$. The agent's objective is to interact with the environment by iteratively calling the tool to gather visual evidence, this process continues until the agent determines it has sufficient information to produce a final answer. The agentic approach enables the model to develop diverse and complex reasoning strategies, as demonstrated in Figure 3. Each tool calling is constrained by a frame budget $B$ (i.e. $f_{high} \times (t_{end} - t_{start}) \leq B$), thus the total number of frames that can be requested from the high-resolution clips is limited by $B \times N$, where $N$ is the maximum number of interaction rounds. The environment returns an error message if the model makes an invalid request or exceeds the frame budget. The goal is to learn a policy $\pi$ that maximizes the quality of the final answer while adhering to the frame budget and tool call number constraints.

## 3.2 COLD-START INITIALIZATION

Reinforcement learning from scratch on a complex, high-dimensional action space, such as generating structured tool calls, is often sample-inefficient and prone to instability. To mitigate these challenges, we precede the RL phase with a supervised fine-tuning (SFT) stage designed to "cold-start" our agent. The primary objective of this stage is twofold: first, to equip a base multimodal model with the fundamental capability of understanding and invoking the `<video_zoom>` tool in the correct format; and second, to expose it to a diverse range of reasoning patterns, which is crucial

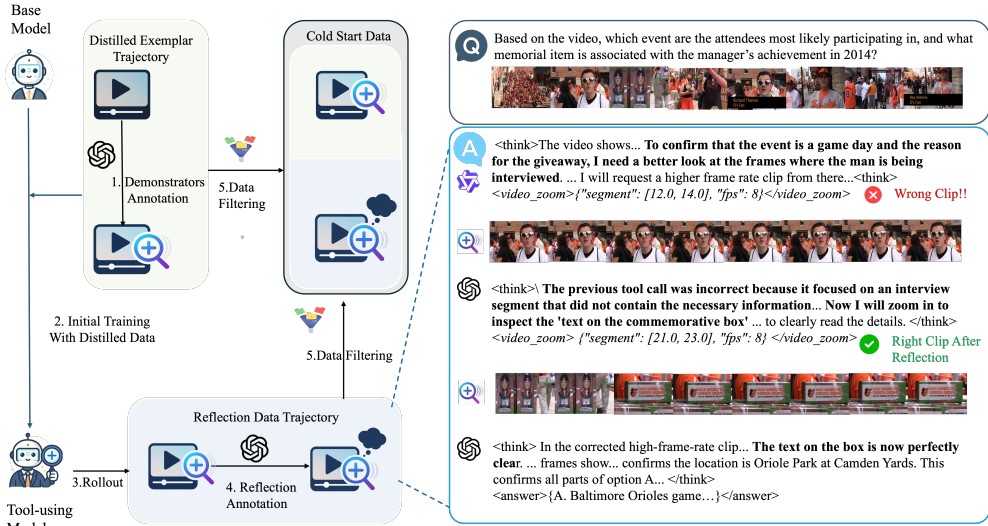

Figure 4: **The pipeline for curating our cold-start dataset.** We first distill exemplar trajectories, then generate reflection data by having an expert model correct the failures of an initial agent. The final dataset combines both verified exemplar and reflection trajectories.

for effective exploration during subsequent RL training. To achieve this, we construct a specialized SFT dataset by curating high-quality, multi-turn interaction trajectories as illustrated in Figure 4.

**Distillation of Exemplar Trajectories.** The initial step is to generate a set of "golden" tool-use trajectories. We leverage state-of-the-art proprietary models, such as GPT-4o (OpenAI, 2024) and Gemini-2.5-pro (Comanici et al., 2025), as expert demonstrators. For each video-question pair in our training set, we prompt the expert model with the same system prompt and initial low-frame-rate video provided to our agent. The model then engages in a multi-turn interaction, iteratively calling the <video_zoom> tool until it gathers sufficient information to answer the question. This process yields a collection of complete trajectories, each containing the initial prompt, a sequence of tool calls, the corresponding high-frame-rate clip observations, and the final answer. These expert-generated trajectories serve as ideal examples of effective tool invocation and reasoning.

**Augmentation with Reflection Data.** While SFT on only exemplar trajectories effectively teaches the model the format of tool use, we observed a significant limitation: the resulting model tends to overfit the expert model's dominant reasoning patterns. This often leads to a "shallow" policy, where the model learns to call the tool at most once and then immediately outputs an answer, regardless of whether the retrieved clip was actually helpful or contained errors. This lack of perseverance and adaptability would severely hinder its ability to solve more complex problems requiring deeper, iterative reasoning.

To address this and introduce more diverse and complex reasoning patterns, we generate reflection data. As shown in Figure 4. The process begins by using our initial model trained only on exemplar data, to produce its own rollouts. We then identify trajectories where the model failed to answer correctly. These incorrect rollouts are subsequently fed back to the expert model, which is prompted to reflect on the flawed reasoning. The model then identifies the mistake and generates a corrected, more robust reasoning path. This corrected path might involve additional tool calls or a different line of reasoning. This reflection process creates valuable training instances that explicitly teach the model how to recover from errors, critically evaluate the information returned by a tool, and when to persist with further investigation. Furthermore, this on-policy-like data generation strategy ensures that the new trajectories are challenging yet achievable, mitigating distribution shift and stabilizing the transition from SFT to RL.

The final cold-start dataset is a carefully curated combination of the distilled exemplar trajectories and reflection trajectories. Before inclusion, all candidate trajectories are passed through verifiers to

ensure quality. This resulted composite dataset, approximately 11,000 trajectories in total, provides a rich and balanced foundation for our base model.

## 3.3 MULTI-TURN TOOL-INTEGRATED REINFORCEMENT LEARNING

We employ GRPO (Shao et al., 2024) for RL training due to its demonstrated efficacy in enhancing multimodal reasoning capabilities, and extend its original formulation to multi-turn tool-calling scenarios by introducing a token-level loss mask over the tool-call trajectory that ignores text and image tokens not generated by the model.

**Reward Design.** The design of the reward function is essential to guide the agent toward the desired behavior. Our reward is assigned at the end of each trajectory and is composed of three distinct components designed to promote accuracy, valid format, and exploration:

$$R(x, y) = R_{acc}(x, y) + R_{format}(y) + R_{tool}(y) \tag{1}$$

The accuracy reward $R_{acc}$ is the primary task-oriented reward, it provides a strong positive signal if the agent's final answer is correct. The format reward $R_{format}$ validates the structure of the agent's response at each turn. This reward is set to a positive value if the model's output strictly adheres to the predefined format, and zero otherwise. Specifically, the agent receives a positive reward if every intermediate step correctly wraps its reasoning in `<think></think>` tags and be followed by either a valid `<video_zoom></video_zoom>` or a final answer enclosed in `<answer></answer>` tags. A key challenge during early training is that a model unfamiliar with the `<video_zoom>` tool may be hesitant to use it, often preferring to guess an answer directly. To solve this and encourage exploration, we introduce a bonus $R_{tool}$ for using the tool. To prevent the agent from learning to make redundant or unhelpful tool calls, this bonus is conditional: it is only awarded if the final answer is correct.

## 4 EXPERIMENT

### 4.1 EXPERIMENTAL SETUP

**Implementation Details.** We initialize our model from Qwen-2.5-VL-7B-Instruct (Bai et al., 2025) for its strong foundational capabilities and amenability to reinforcement learning. For cold-start initialization, we adapt the LLaMA-Factory (Zheng et al., 2024) framework. Our RL training and evaluation framework is based on verl (Sheng et al., 2024), which we extended to support multi-turn tool-calling tasks and optimized for efficiency in long video training scenario.

For training data, we use LongVideoReason (Chen et al., 2025), a long video QA dataset comprised of 52K high-quality question-reasoning-answer pairs. In cold start stage, we trained our base model with a learning rate of $5 \times 10^{-6}$ for 1 epoch on dataset we construct as described in Section 3.2. During RL stage, we use a learning rate of $1 \times 10^{-6}$, rollout number of 16 and batchsize of 128. The model is initialized with 64 uniformly sampled frames. It can then perform up to 4 subsequent tool calls, each retrieving up to 16 frames of high-resolution clip from a segment of interest, before providing a final answer. To improve training effectiveness and stability of RL training process, we also adapt clip-higher and dynamic sampling from DAPO (Yu et al., 2025). Further details are provided in the appendix.

**Benchmarks.** To comprehensively evaluate the capabilities of our model, we conducted tests on two distinct categories of benchmarks: long video understanding and long video reasoning. For long video understanding, we utilized four benchmarks: MLVU (Zhou et al., 2024), LongVideoBench (Wu et al., 2024), VideoMME (Fu et al., 2024), and LVBench (Wang et al., 2024a). These benchmarks encompass a variety of tasks designed to assess the model's general video comprehension abilities. For long video reasoning, we employed three benchmarks that require more than superficial visual analysis: VideoMMLU (Song et al., 2025), VideoMMMU (Hu et al., 2025b), and LongVideoReason-eval (Chen et al., 2025). These chanllenging benchmarks are specifically designed to evaluate the model's integrated perception and reasoning capabilities.

Table 1: **Results on long video benchmarks**.[†] denotes evaluation results using our own evaluation protocol under max frames of 128. For a fair comparison, our model is evaluated with a maximum of 64 frames in the first round, followed by up to 4 turns requesting a maximum of 16 frames per turn, yielding a total of max 128 frames.

| Model | Size | Long Video Understanding | | | | | | | Long Video Reasoning | | |
|---|---|---|---|---|---|---|---|---|---|---|---|
| | | MLVU | | LongVideoBench | VideoMME | | LVBench | | VideoMMLU | VideoMMMU | LongVideoReason |
| | | dev | test | val | overall | long | | | quiz | | eval |
| *Proprietary Models* | | | | | | | | | | | |
| GPT-4o | - | 64.6 | 54.9 | 66.7 | 71.9 | 65.3 | 48.9 | | 44.9 | 61.2 | 60.7 |
| Gemini-1.5-Pro | - | - | - | 64.0 | 75.0 | 67.4 | 33.1 | | - | 53.9 | 67.3 |
| *Open-Source VLMs* | | | | | | | | | | | |
| Video-LLaVA | 7B | 36.2 | 30.7 | 37.6 | 39.9 | - | - | | - | - | - |
| LLaVA-OneVision | 7B | 64.7 | 47.2 | 56.4 | 58.3 | 46.7 | - | | 33.4 | 33.9 | - |
| LLaVA-NeXT-Video | 7B | - | - | 49.1 | - | - | - | | 27.6 | - | - |
| Video-XL | 7B | 64.9 | 45.5 | 50.7 | 55.5 | - | - | | - | - | - |
| VILA-1.5 | 7B | 56.7 | - | - | - | - | - | | 20.5 | 20.9 | - |
| Kangaroo | 8B | 61.0 | - | 54.8 | 56.0 | - | 39.4 | | - | - | - |
| LongVU | 7B | 65.4 | - | - | 60.6 | - | - | | - | - | - |
| LongVA | 7B | 56.3 | 41.1 | - | 52.6 | - | - | | - | 24.0 | - |
| LongVILA | 7B | - | - | 57.1 | 60.1 | - | - | | - | - | - |
| LongVILA-R1 | 7B | - | - | 57.6 | 62.4 | 53.3 | - | | - | - | 67.9 |
| Video-R1[†] | 7B | 65.0 | 49.2 | 52.0 | 61.1 | 51.4 | 38.7 | | 61.3 | 49.8 | 72.8 |
| Qwen2.5-VL[†] | 7B | 58.3 | 45.5 | 51.0 | 63.5 | 53.9 | 36.9 | | 61.0 | 48.1 | 70.8 |
| VideoZoomer[†] | 7B | **68.8** | **55.8** | **57.7** | **65.2** | **55.8** | **41.5** | | **67.9** | **52.2** | **80.3** |
| *Δ over base model* | | +10.5 | +10.3 | +6.7 | +1.7 | +1.9 | +4.6 | | +6.9 | +4.1 | +9.5 |

Table 2: **Detailed result on MLVU.** ER: Ego Reasoning. NQA: Needle QA, PQA: Plot QA, SQA: Sport QA, AO: Action Order, AC: Action Count, TQA: Tutorial QA, AR: Anomaly Recognition, TR: Topic Reasoning.

| Split | Model | Single Detail | | | | Multi-detail | | | Holistic | | |
|---|---|---|---|---|---|---|---|---|---|---|---|
| | | ER | NQA | PQA | SQA | AO | AC | TQA | AR | TR | Avg. |
| Dev | Qwen2.5-VL | 47.7 | 65.1 | 65.9 | - | 50.2 | 13.6 | - | **65.5** | **85.6** | 58.3 |
| | VideoZoomer | **66.8** | **80.3** | **72.9** | - | **59.8** | **50.5** | - | 52.5 | 83.3 | **68.8** |
| Test | Qwen2.5-VL | 32.1 | 53.3 | 54.0 | **44.4** | 32.9 | 15.0 | 37.2 | 38.5 | 80.2 | 45.5 |
| | VideoZoomer | **58.5** | **63.3** | **64.0** | **44.4** | **42.9** | **28.3** | **39.5** | **46.2** | **89.0** | **55.8** |

## 4.2 MAIN RESULT

**Baselines.** We compare VideoZoomer against a wide range of video understanding models, including (1) Proprietary models: GPT-4o (OpenAI, 2024) and Gemini-1.5-Pro (Reid et al., 2024); (2) Open-source VLMs: Video-LLaVA (Lin et al., 2023), LLaVA-NeXT-Video (Zhang et al., 2024b), Video-XL (Shu et al., 2024), VILA-1.5 (Lin et al., 2024), Kangaroo (Liu et al., 2024a), LongVU (Shen et al., 2024), LongVA (Zhang et al., 2024a), LongVILA (Xue et al., 2024), LongVILA-R1 (Chen et al., 2025), Video-R1 (Feng et al., 2025) and Qwen2.5-VL (Bai et al., 2025).

**Long Video Understanding.** Our model demonstrates marked improvements across a range of long video understanding benchmarks, as shown in Table 1. On MLVU, it achieves scores of 66.8 (dev) and 55.8 (test), yielding substantial gains of +10.5 and +10.3 points over its base model, Qwen2.5-VL. This performance is further validated on LongVideoBench and LVBench, where our model scores 57.7 and 41.5, respectively, outperforming all listed open-source baselines. These results collectively underscore the effectiveness of our adaptive temporal zoom mechanism. Notably, even on benchmarks not exclusively focused on extremely long durations like VideoMME, our method provides a clear performance boost (65.2 overall, 55.8 on long-set) over an already strong baseline. This demonstrates that the learned policy to dynamically "zoom" in relevant segments is beneficial across various video length.

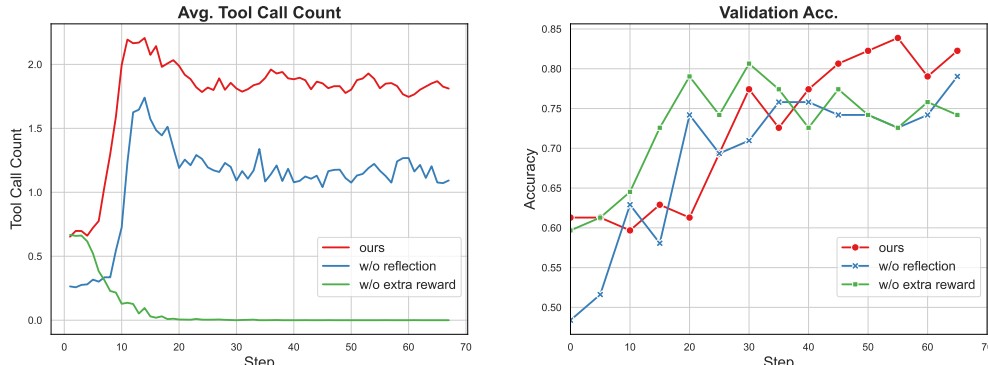

Figure 5: **Training dynamics of ablation baselines.** The left panel shows the average number of tool calls per sample during training. The right panel displays the model performance (e.g., accuracy) on the validation set over the course of training.

Table 3: **Evaluation of Ablation Baselines**.

| Model | Long Video Understanding | | | | | | Long Video Reasoning | | |
|---|---|---|---|---|---|---|---|---|---|
| | MLVU | | LongVideoBench | VideoMME | | LVBench | VideoMMLU | VideoMMMU | LongVideoReason |
| | dev | test | val | overall | long | | quiz | | eval |
| VideoZoomer | **68.8** | **55.8** | **57.7** | **65.2** | **55.8** | **41.5** | 67.9 | 52.2 | **80.3** |
| w/o RL | 56.4 | 45.6 | 42.0 | 54.4 | 44.2 | 26.0 | 63.5 | 46.6 | 63.3 |
| w/o $R_{tool}$ | 67.5 | 52.2 | 56.2 | 62.5 | 52.5 | 40.6 | 63.6 | **53.8** | 79.9 |
| w/o cold-start | 57.0 | 42.8 | 43.5 | 53.5 | 46.6 | 35.5 | 63.9 | 43.6 | 59.6 |
| w/o reflection | 67.0 | 53.2 | 54.8 | 58.7 | 47.4 | 40.9 | 70.1 | 52.2 | 75.1 |

We present a detailed analysis of our model's performance on the MLVU benchmark in Table 2. The results clearly show that our method's improvements are most significant on tasks requiring detailed perception. For instance, in the "Single Detail" category of the dev set, our model shows massive gains in Ego Reasoning (ER, +19.1), Needle QA (NQA, +15.2), and Plot QA (PQA, +7.0). The most significant improvement is seen in the "Multi-detail" task of Action Count (AC), where our model's score increases from 13.6 to 50.5. This task, which requires counting specific, often rapid actions, directly benefits from the ability to re-sample critical moments at a higher frame rate. Similar substantial gains are observed on the test set in ER (+26.4), NQA (+10.0), and AC (+13.3).

**Long Video Reasoning.** On VideoMMLU and VideoMMMU, our model scores 67.9 and 52.2 respectively, achieving the highest among all open-source models. On LongVideoReason-eval, our model achieves a highest score of 80.3, surpassing the performance of powerful proprietary models like GPT-4o (60.7) and Gemini-1.5-Pro (67.3). Notably, our model also outperforms LongVILA-R1, which is trained on the same dataset but with a larger frame budget, highlighting the superior efficiency of our agentic strategy. This indicates that the iterative, evidence-gathering process enabled by our agentic strategy allows the model to construct more robust and accurate reasoning chain, which is crucial for tackling complex, knowledge intensive video reasoning tasks.

## 4.3 ABLATION STUDY

**Effectiveness of Key Components.** To validate the contribution of each key component in our framework, we conduct a comprehensive ablation study, with results summarized in Table 3 and training dynamics shown in Figure 5. For a fair comparison, all ablated models except "w/o cold-start" were trained using the same amount of SFT data as our final model. The w/o RL model, trained only via supervised fine-tuning, suffers a catastrophic performance drop across all benchmarks (e.g., -17.0 on LongVideoReason), confirming that RL is essential for learning an effective

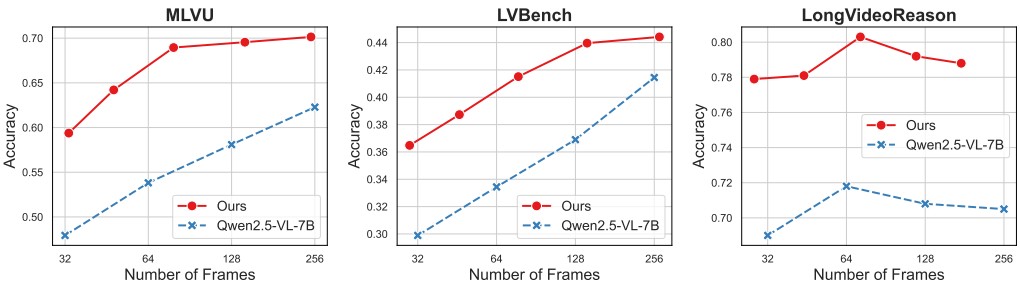

Figure 6: **Performance comparison with varying frame budgets.** We compare our model against the Qwen2.5-VL-7B baseline. The x-axis(log scale) represents the fixed frame budget for the baseline and the average number of frames actually used by VideoZoomer on each benchmark.

tool-use policy. Similarly, the w/o cold-start model, which skips our curated SFT stage, fails to converge to a meaningful policy, highlighting the necessity of a strong initialization. Within the cold-start process, removing reflection data (w/o reflection) causes the model to adopt a shallow, simple strategy, where the average tool call count stabilizes at about 1.0, limiting its ability to tackle complex problems. In contrast, our full method learns to make nearly two calls on average, enabling deeper investigation and achieving higher accuracy in the validation set. Finally, removing the conditional tool-use bonus (w/o $R_{tool}$) leads to "policy collapse", where the agent's tool usage trends towards zero during training, as it fails to discover the tool's utility without explicit encouragement. Each ablation results in significantly lower performance on various benchmarks, demonstrating that all components are necessary to achieve the final performance.

**Performance Across Various Frame Budgets.** To further investigate the efficiency of Video-Zoomer, we analyze the performance of our model and the base model under various frame budgets. As illustrated in Figure 6, we plot the accuracy of the model against the number of frames processed. For the baseline model, this x-axis represents a fixed, uniformly sampled frame budget. For our model, it represents the actual average number of frames consumed per dataset, a result of its dynamic decision-making process. The results clearly demonstrate our model's superior efficiency. On MLVU, our model achieves 0.64 accuracy using only 48 frames on average, surpassing the baseline's 0.581 accuracy at a much larger 128 frame budget. This trend holds on LVBench, where our model using 77 frames outperforms the baseline using 256 frames. Furthermore, on the LongVideoReason benchmark, our model and the baseline model both peaks at around 64 frames, suggesting that complex reasoning tasks may not benefit from increasing visual information, which can introduce noise. However, within this optimal frame window, our model's peak accuracy of 0.803 significantly surpasses the base model's peak of 0.718. This performance gap underscores our model's stronger reasoning capability enabled by its agentic policy.

**Combining with a Frame Selector.** Our primary method uses uniform sampling for the initial overview to ensure a global and unbiased starting point, we also investigate whether our agentic framework can be combined with more sophisticated frame selectors. To test this, we replace the initial uniformly sampled frames with the output of the output of TSPO-0.4B (Tang et al., 2025). The results presented in Table 4 shows that providing a more intelligently selected initial overview further boosts our model's performance by +2.0 on MLVU

Table 4: **Performance comparison when combining with an external frame selector.** Results are evaluated using our protocol under a consistent setting.

| Model | MLVU | LongVideoBench |
|---|---|---|
| Qwen2.5VL | 58.1 | 51.0 |
| +*tspo* | 68.1 | 54.9 |
| VideoZoomer | 68.8 | 57.7 |
| +*tspo* | **70.8** | **60.7** |

and +3.0 on LongVideoBench. This demonstrates the flexibility and transferability of our approach; the learned policy effectively leverages the improved starting point to conduct an even more efficient and accurate investigation of the video.

# 5 CONCLUSION

In this work, we propose VideoZoomer to address the critical challenge of long video understanding in MLLMs. We empower the MLLM to become an active agent capable of utilizing external tool to investigate long videos more effectively and efficiently through a carefully designed two-stage training process.

Our experimental results robustly validated our approach. The ablation studies confirmed that each component—the cold-start initialization, the reflection data, the RL optimization, and the conditional reward bonus—was indispensable for achieving final performance. Our model not only achieves strong performance across numerous long video benchmarks, but also demonstrated superior frame efficiency, outperforming stronger baselines while using significantly fewer frames. This demonstrates the effectiveness of our agentic strategy in enhancing the perception and reasoning capabilities of MLLMs for long videos.

## ACKNOWLEDGMENTS

This work was partly supported by the National Natural Science Foundation of China (Grant No. 62576191) and the Shenzhen Science and Technology Program (ZDCY20250901103533010) .

## ETHICS STATEMENT

Our research adheres to the ICLR Code of Ethics. This work aims to advance the efficiency of video understanding for positive applications, ensuring transparency, reproducibility, and fairness in all experiments. All datasets used are publicly available, and our use of proprietary models for data distillation complies with their terms of service. We acknowledge that our model, like other large language models, may inherit and reflect biases from its training data. While our method improves reasoning, it is not designed to mitigate social biases, and we advocate for responsible downstream use and further research into fairness. The intended application is for research purposes, and we do not foresee direct dual-use concerns from our proposed framework.

## REPRODUCIBILITY STATEMENT

To ensure the reproducibility of our work, we provide a comprehensive overview of our methodology, implementation, and evaluation. Further implementation details, including training hyperparameters and specific prompts used for training and data generation, are described in Appendix A. To facilitate direct replication and further research, we will release our codebase, datasets and model weight upon acceptance of this paper.

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

# A    MORE IMPLEMENT DETAILS

**Evaluation Details.**   We evaluate our model and baselines under a consistent setting with a maximum of 128 frames and a resolution corresponding to 100,352 pixels per frame. For inference, we employed the vLLM framework (Kwon et al., 2023) with the temperature parameter set to 0 to ensure deterministic outputs.

For the VideoMMLU benchmark, answers are scored by GPT-4o using the official prompt and the final score is computed as the average score of three disciplines.

**Training Details.**   We show the key training hyperparameters in Table 5.

Table 5: Key Hyperparameters

(a) SFT stage

| Hyperparameter | Value |
|---|---|
| Train epochs | 1 |
| Train batch size | 64 |
| Learning rate | 5e-5 |
| Learning rate scheluder | cosine |
| Warmup ratio | 0.1 |
| Freeze vision encoder | true |

(b) RL stage

| Hyperparameter | Value |
|---|---|
| Max total response length | 32768 |
| Rollout temperature | 1.0 |
| Max interaction turns | 5 |
| Train batch size | 128 |
| PPO mini batch size | 32 |
| Rollouts per prompt ($n$) | 16 |
| Clip ratio (low / high) | 0.2 / 0.27 |
| Entropy coefficient | 0.001 |
| KL coefficient ($\beta$) | 0.001 |
| Learning rate | 1e-6 |
| Reward weight (acc/format/tool) | 0.9/0.1/0.5 |

The SFT training is conducted on 8×H100 GPUs for ∼6h, RL training is conducted on 16×H100 GPUs for ∼45h.

Figure 7 shows key statistics of our cold-start dataset. The left panel shows the distribution of total token lengths per trajectory, indicating a wide variety of response lengths that cover both simple and complex reasoning chains. The right panel illustrates the distribution of interaction rounds (i.e., the number of tool calls), showing that the dataset contains a significant number of multi-step examples.

**Prompt Template.**   We provide the detailed prompt (Ma et al.; Zhang et al., 2025a) used for training and cold-start data synthesization as follows:

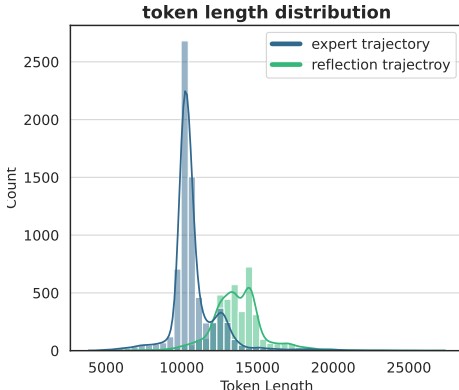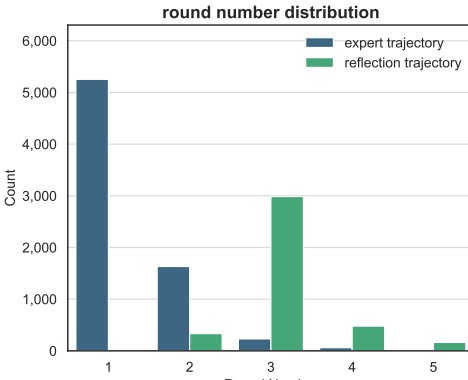

Figure 7: Statistics of the cold start dataset.

---

### Reasoning Prompt

**System Prompt:** You are a helpful assistant. You will receive a low-frame-rate video and related questions. You can analyze the video content to answer the question and trigger high-frame-rate inspections when finer temporal resolution is needed. When you detect ambiguous motion/objects that require closer inspection, wrap your request in `<video_zoom></video_zoom>` tags and provide the exact time segment and target frame rate in JSON format: `<video_zoom>{"segment": [start_sec, end_sec], "fps": n} </video_zoom>`, it will return the video clip at the target fps to help you better answer the question. Note that the total frames num of the request clip cannot exceed 16 (e.g. (end_sec - start_sec) * fps ≤ 16) and DO NOT include `<answer>` tags in this round. Example usage: `<video_zoom> {"segment": [4.0, 6.0], "fps": 2} </video_zoom>`. If the initial tool response does not provide sufficient information to answer the question, you may continue to request additional video zoom inspections as needed, until you either (1) gather enough information to form a complete answer, or (2) are explicitly instructed to stop using the tool. Output the thinking process within `<think> </think>` tags, once you confirm your final answer place the final answer inside `<answer>` and `</answer>`.

**User:** ...`<framei_timet><image>`...Question

---

### Prompt for Constructing Reflection Trajectory

You are an expert video understanding model with access to a video zoom tool that allows you to request high-frame-rate clips for temporal inspection. Your task is to correct a flawed analysis of a low-frame-rate video by using a `video_zoom` tool. Your workflow is a multi-turn process:

**Turn 1: Reflection and Tool Call**

1. **Analyze the Error**: You will be given a question, choices, and a previous, incorrect attempt. First, you must reflect on *why* the previous `video_zoom` tool call was flawed. Was the time segment wrong? Was the frames-per-second (fps) too low? Was the focus of the analysis misaligned with the question?

2. **Formulate a Correction**: Based on your analysis, decide on a new, corrected `video_zoom` request. This request should target the precise moment of interest and use an appropriate fps to capture the fine-grained detail.

3. **Output the Tool Call**: Generate your reflection and the new tool call in the specified format. **Your output for this turn MUST end immediately after the `</video_zoom>` tag.** Do not generate anything further. The system will then execute this call and provide you with the result.

**Constraint for the tool call**: The total number of frames requested must not exceed 16. That is: `(end_sec - start_sec) * fps <= 16`.

**Turn 2: Analysis and Final Answer**

1. **Receive Tool Response**: The system will provide the high-frame-rate video clip from your corrected tool call.

2. **Analyze the New Clip**: Carefully examine the new clip. Describe what you can now clearly see that resolves the question.

3. **Provide the Final Answer**: Based on your new observation, state the correct answer from the choices, enclosed in `\boxed{}`.

**Output Format Structure:**
**[FIRST TURN OUTPUT]**
`<think>`
The previous tool call was incorrect because [explain the flaw in the tool use, e.g., wrong segment, wrong fps, or misaligned focus].
Now I will zoom in to inspect the motion of '{target object/action}' between {start_sec}s and {end_sec}s with higher temporal resolution.
`</think><video_zoom> {"segment": [start_sec, end_sec], "fps": n} </video_zoom>`
**[YOUR TURN 1 OUTPUT STOPS HERE]**
**[SECOND TURN OUTPUT]** (after you receive the tool response)
`<think>` In the corrected high-frame-rate clip, [describe what is clearly observed now]. `</think>`
`<answer> \\boxed{correct answer} </answer>`
**Example to follow:**

---

> **Question:** Which hand did the woman use to pick up the cup?
> **Choices:** A: Left hand B: Right hand C: Both hands D: Neither
> **Previous Trajectory (Wrong):** Tool call: `<video_zoom> {"segment": [0.0, 2.0], "fps": 2}</video_zoom>`
> **(Your First Turn Output Should Look Like This):**
> `<think>` The previous tool call was incorrect because it focused on the wrong time segment. The woman only reaches for the cup between 3.0s and 5.0s. Additionally, the low fps of 2 might not be sufficient to clearly distinguish the hand's motion.
> Now I will zoom in to inspect the motion of 'the woman's hand reaching for the cup' between 3.0s and 5.0s with a higher temporal resolution. `</think><video_zoom> {"segment": [3.0, 10.0], "fps": 1} </video_zoom>`
> **(System provides tool response, then you start your Second Turn) (Your Second Turn Output Should Look Like This):**
> `<think>`In the corrected high-frame-rate clip, the woman's right hand is clearly seen moving towards and gripping the cup handle between 4.1s and 4.8s, while her left hand remains on her lap. The motion is now unambiguous. `</think> <answer>` B. `</answer>`

## B    MORE EXPERIMENT RESULTS

### B.1    DIFFERENT EXPERT MODEL FOR COLD-START DATA CONSTRUCTION.

In our main experiments, we utilized data distilled from Gemini 2.5 Pro (Comanici et al., 2025) to generate the cold-start SFT dataset. To justift this choice, we conduct a comparative analysis of data distilled from Gemini 2.5 pro versus data from GPT-4o. We conducted two identical training runs of our model. The only difference was the source of the cold start dataset used in the SFT stage: one model was trained on data distilled from GPT-4o, and the other on data from Gemini 2.5 Pro. Both models then underwent the same reinforcement learning phase. We evaluated the final performance of both models on our key benchmarks. The results of this comparison are presented in Table 6. As the result shows, the model trained using data from Gemini 2.5 pro achieved slightly better results on most benchmarks. Through qualitative analysis of the generated data, we observed that the trajectories from Gemini exhibited greater diversity in their reasoning patterns and tool-use strategies.

Table 6: Performance comparison using different expert models for cold-start data construction.

| Model | Size | MLVU | | LongVideoBench | VideoMME | | LVBench | VideoMMLU | VideoMMMU | LongVideoReason |
|---|---|---|---|---|---|---|---|---|---|---|
| | | dev | test | val | overall | long | | quiz | | eval |
| Qwen2.5-VL | 7B | 58.1 | 45.4 | 51.0 | 63.5 | 53.9 | 36.9 | 61.0 | 48.1 | 70.8 |
| VideoZoomer$_{gemini}$ | 7B | 68.8 | **55.8** | **57.7** | **65.2** | **55.8** | **41.5** | **67.9** | **52.2** | **80.3** |
| VideoZoomer$_{gpt-4o}$ | 7B | **69.5** | 54.6 | 55.5 | 61.6 | 51.0 | 41.2 | 64.1 | 51.2 | 78.4 |

### B.2    RESULTS ON OOD TASKS

To assess the generalizability and robustness of VideoZoomer, we evaluated its performance on two distinct out-of-distribution (OOD) task categories: short video captioning and logical reasoning on synthetic data. These experiments were designed to verify that our training process enhances long-video capabilities without degrading the model's foundational abilities.

While our primary focus is on long videos, we tested VideoZoomeron several short video captioning benchmarks TemporalBench(Cai et al., 2024),TempCompass(Liu et al., 2024b) and VDC(Chai et al., 2024) to ensure its core descriptive capabilities were maintained. The results, summarized in Table 7, show that our model not only preserves but significantly improves upon the baseline's performance across all tested benchmarks.

We further tested the model's robustness (Li et al., 2021) on a subset of the CLEVRER dataset(Yi et al., 2019), which evaluates causal and logical reasoning on synthetic videos. This domain is significantly different from the real-world (Kang et al., 2025; Li et al., 2025), long-form videos used in our training.

Table 7: Short Video Captioning Benchmark Results

| Model | TemporalBench (Short Caption Score) | TempCompass (Captioning Acc) | VDC (Short Acc / Score) |
|---|---|---|---|
| QwenVL-2.5-7B | 40.9 | 52.1 | 37.8 / 1.98 |
| VideoZoomer | 56.4 | 65.3 | 49.2 / 2.51 |

As shown in Table 8, the comparable performance to the baseline model demonstrates that our two-stage training process does not degrade the model's foundational reasoning abilities. The minimal gain is expected, as the glance-and-zoom mechanism is not designed for the abstract, logical puzzles presented by CLEVRER. This result confirms that our method successfully retains the model's core competencies on tasks that do not require our agentic framework.

Table 8: Performance on CLEVRER

| Model | CLEVRER Accuracy |
|---|---|
| QwenVL-2.5-7B | 67.3 |
| VideoZoomer | 68.0 |

## B.3 IMPACT OF SFT DATA QUANTITY

We investigated whether the effectiveness of our Supervised Fine-Tuning (SFT) phase stems from the quantity of data. We compared our model, trained on our curated 11k trajectory dataset, against a model trained on a dataset of the same composition but with double the quantity (∼20k samples).

Table 9: Impact of SFT Data Quantity

| Training Dataset | MLVU (dev) | MLVU (test) | LVBench | LongVideoBench | LongVideoReason-eval |
|---|---|---|---|---|---|
| Ours (∼11k) | 68.8 | 55.8 | 41.5 | 57.7 | 80.3 |
| Scaled Dataset (∼20k) | 66.4 | 56.0 | 41.4 | 55.6 | 80.3 |

As shown in Table 9, simply doubling the data quantity did not lead to better overall performance. While there was a marginal improvement on MLVU (dev), the model trained on the larger dataset performed worse on all other benchmarks. This result strongly suggests that the effectiveness of our dataset comes from the high-quality, diverse reasoning patterns it contains, rather than its sheer size. This "less is more" philosophy aligns with recent findings from works like DeepSeek-R1(Guo et al., 2025) and LIMO(Ye et al., 2025), which demonstrate that a few thousand high-quality, reasoning-focused samples can be sufficient to unlock powerful capabilities in large models. Our methodology prioritizes a rich collection of reasoning pathways over a large volume of repetitive examples. (Li et al., 2024)

## B.4 ANALYSIS OF CHOSEN FPS

A key feature of our `<video_zoom>` tool is that the frames-per-second (fps) for a "zoom-in" clip is dynamically generated by the model itself, allowing it to decide not only where to look but also how closely to look. To understand the model's learned behavior, we analyzed the distribution of fps values it chose across thousands of tool calls on our validation set.

The results in Table 10 reveal that the model does not default to the highest possible fps. Instead, its most frequent choice is a moderate fps in the (1, 2] range, which it selects in 66.2% of cases. This demonstrates that the model learns an efficient policy, requesting just enough temporal detail to solve the task without unnecessarily expending its frame budget. While a high fps like 8 might seem

excessive for a full video, it is a reasonable and effective choice for examining a critical few-second clip, and the model learns to use it sparingly.

Table 10: Distribution of fps Values Chosen by the Model

| fps Range | (0, 1] | (1, 2] | (2, 4] | (4, 8] | (8, ∞) |
|---|---|---|---|---|---|
| Percentage | 24.9% | 66.2% | 8.2% | 0.6% | ≤0.1% |

### B.5 PERFORMANCE BY MAXIMUM ALLOWED TOOL CALLS

To understand the impact of multi-step reasoning, we evaluated how model accuracy changes with the maximum number of allowed tool calls. We varied the limit from 0 (no tool use) to 4 and measured performance across several benchmarks.

Table 11: Performance by Maximum Allowed Tool Calls

| Max Tool Calls | MLVU (dev) | MLVU (test) | LongVideoBench | LVBench | LongVideoReason-eval |
|---|---|---|---|---|---|
| 0 (No Tool Use) | 65.0 | 52.6 | 52.7 | 38.7 | 64.8 |
| 1 | 66.7 | 53.4 | 55.3 | 39.6 | 79.9 |
| 2 | 67.8 | 56.6 | 57.0 | 41.4 | 80.2 |
| 3 | 68.4 | 56.0 | 56.2 | 41.8 | 81.1 |
| 4 | 68.8 | 55.8 | 57.7 | 41.5 | 80.3 |

The results in Table 11 show a clear trend. The most significant performance jump occurs when allowing just one tool call, confirming the fundamental value of our agentic framework. For example, on LongVideoReason-eval, accuracy leaps from 64.8% to 79.9%. Increasing the limit from one to two calls provides another consistent and significant boost, highlighting the importance of multi-turn interaction for refining searches or gathering more evidence. After two calls, the benefits of increasing the limit further show diminishing returns. While there are minor gains, the improvements are smaller and less consistent. Our final model, with a limit of 4, uses this extra budget sparingly for the hard problems, but the core performance is achieved within the first two interactions.

## C MORE CASES

Q:How many green mugs were on the fridge top after I first opened the fridge?A. 0 B. 2 C. 3 D. 1

<think>The user is asking about the number of green mugs on the fridge top after the first time the fridge is opened. The relevant frames are around **342.83s.** I need to see the motion of the person opening the fridge and what is inside. I will request a higher frame rate clip from 342.0s to 343.0s.</think>
<video_zoom>{"segment": [342.0, 343.0], "fps": 8}</video_zoom>

<think>The zoomed clip clearly shows the person opening the fridge. Inside the fridge, there are two green mugs visible on the top shelf. This confirms that the number of green mugs on the fridge top after the first time the fridge is opened is 2. </think>
<answer>B. 2</answer>

Figure 8: Case Study: Single-Detail Reasoning.

Q:What does the man do before frying the food at the end of the video?
A. Adjusts the kitchen timer
B. Cuts some vegetables
C. Cleans the frying pan
D. Melts butter
E. Sprinkles some pepper on the food
F. Turns on the range hood

<think>The question is about the man's actions before frying the food. The relevant part of the video is towards the end, around 1640s. I need to see the sequence of events leading up to the frying to determine what he does first. I will request a higher frame rate clip for the segment from 1630s to 1640s.</think><video_zoom>{\"segment\": [1630.0, 1640.0], \"fps\": 1}</video_zoom>

<think>In the high-frame-rate clip, I can see the sequence of events clearly.- From 1630s to 1634s, the man is using a whisk to stir something in a pot.- At 1635s, he stops stirring and walks to the stove.- At 1638s, he picks up a stick of butter from a wooden bowl. At 1640s, he is seen adding the butter to a pan on the stove. This confirms that before he fries the food, he melts butter in the pan. The other options are not shown in this specific sequence.</think><answer>D</answer>

Figure 9: Case Study: Single-Detail Reasoning.

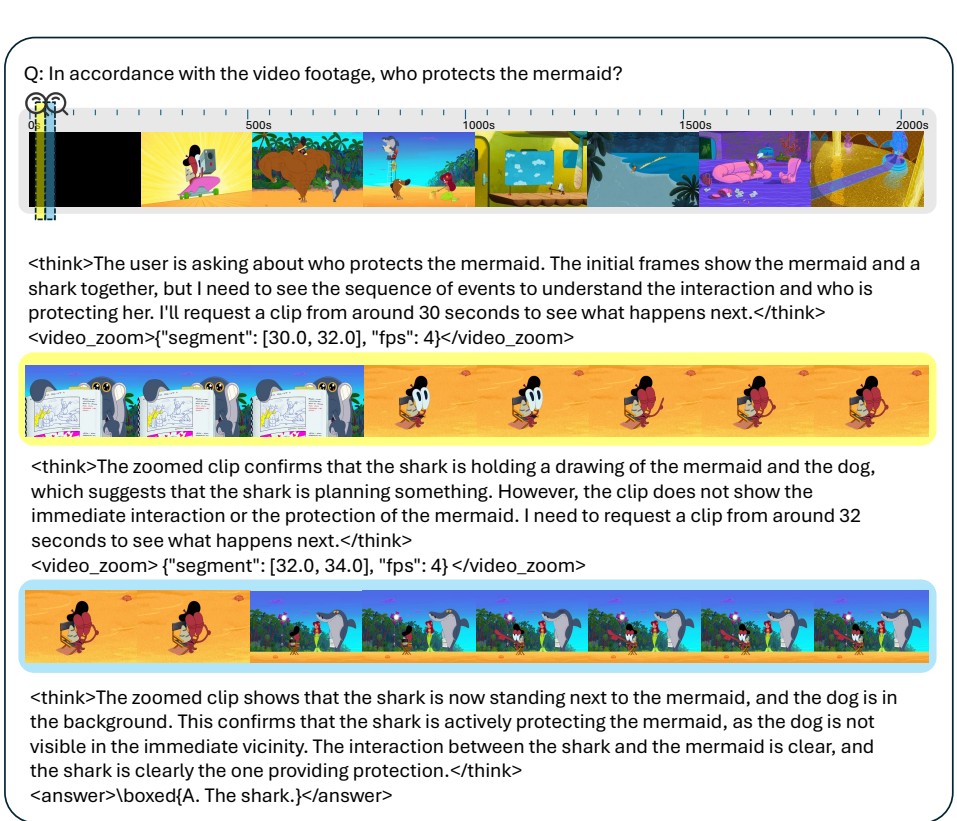

Figure 10: Case Study: Self-Correction and Refined Reasoning.

Q: Which of the following options correctly matches the sequence of actions as they actually appear in the video?
A. making jewelry --> cleaning toilet --> paragliding --> carving pumpkin
B. paragliding --> making jewelry --> cleaning toilet --> carving pumpkin
C. carving pumpkin --> cleaning toilet --> making jewelry --> paragliding
D. paragliding --> cleaning toilet --> carving pumpkin --> making jewelry

<think>The user wants me to match the sequence of events presented in different time clips provided in the video frames. To accurately match the sequence, I need to clarify the order of specific moments mentioned in the question. A higher frame rate inspection around the relevant timestamps is necessary to get precise details. I'll start with the segment where a paraglider is visible.</think>
<video_zoom>{"segment": [50.0, 52.0], "fps": 8}</video_zoom>

<think>The first clip confirmed that the sequence includes paragliding. Next, I need to confirm the order of the other specified activities\u2014making jewelry, cleaning a toilet, and carving pumpkins. I'll zoom into another section where jewelry-making and cleaning the toilet are visible for better details and context. I'll focus on the relevant timestamps for these activities.\n</think>
<video_zoom> {\"segment\": [90.0, 92.0], \"fps\": 4} </video_zoom>

<think>The second clip confirms that the activity of making jewelry occurs before cleaning the toilet. Now, I will inspect the segment where cleaning a toilet is visible to better understand the sequence of these events. The cleaning clips usually appear towards the later part of the sequence
 I will request a clip from around 180.0s to 182.0s to see the transition from the previous scene to the next.</think><video_zoom> {"segment": [180.0, 182.0], "fps": 8} </video_zoom>

<think>The third clip confirms that the cleaning of the toilet follows the \u201cmaking jewelry\u201d activity. Finally, I need to confirm the last action, which is carving a pumpkin. The pumpkin carving is at the initial stages in the video sequence. I will now request a sequence around the moment this specific activity is visible in greater detail.<think><video_zoom> {"segment": [279.0, 281.0], "fps": 8} </video_zoom>

<think>The zoomed clip confirms that carving the pumpkin is shown between 279.0s and 281.0s. This confirms that carving the pumpkin is the final event in the sequence. Combining all the observations, the correct sequence of events is paragliding, making jewelry, cleaning the toilet, and carving the pumpkin.</think>
<answer>B. paragliding --> making jewelry --> cleaning toilet --> carving pumpkin</answer>

Figure 11: Case Study: Multi-Detail Reasoning for Complex Queries.

Based on the video, which event are the attendees most likely participating in, and what commemorative item is associated with the manager\u2019s 2014 achievement?
A. A Baltimore Orioles home game at Camden Yards, featuring a 2014 AL Manager of the Year garden gnome giveaway
B. A Baltimore Ravens playoff game, distributing Super Bowl commemorative footballs
C. A Baltimore Orioles fan convention, offering signed baseballs by 2014 MVP players
D. A charity event at a local park, giving away team jerseys from the 2014 season  \n\n"

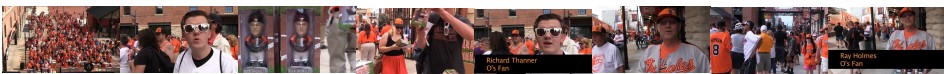

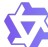 <think>The video shows a crowd at what appears to be a baseball stadium (**\"Camden Yards\" is visible**), primarily wearing orange \"Orioles\" shirts. Several frames show people handing out and presenting commemorative boxes. In the background, we see \"Buck Showalter\" on products. **Frame 11 confirms a \"Buck Showalter Garden Gnome 2014 AL Manager of the Year.\"** The event is clearly associated with the Baltimore Orioles. To confirm that the event is a game day and the distribution is specifically for the manager's 2014 achievement, I need a better look at the frames where the man is being interviewed. The\u5165\u624b at timestamp 12s seems most relevant. **I will request a higher frame rate clip from there** to see the specific moment when the product is presented and the reason for the giveaway is mentioned.</think>
<video_zoom>{\"segment\": [12.0, 14.0], \"fps\": 8}</video_zoom>

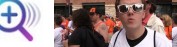 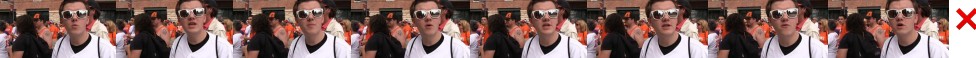

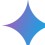 <think>\nThe previous tool call was incorrect because it focused on an interview segment that did not contain the necessary information to answer the question. The question requires identifying both the event and the specific commemorative item. While the video generally shows an Orioles event, **the previous call from 12s-14s only shows a fan talking and provides no details about the item itself. The crucial information is presented on the boxes being given away.** Frame 11 at 22.02s and frame 12 at 24.02s clearly show the item's box.\nNow I will zoom in to inspect the 'text on the commemorative box' between 21.0s and 23.0s with a higher temporal resolution to clearly read the details.\n</think>
<video_zoom> {\"segment\": [21.0, 23.0], \"fps\": 8} </video_zoom>

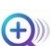 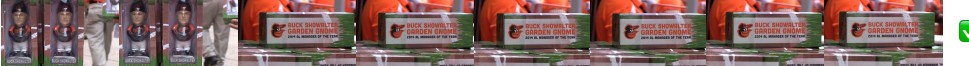

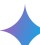 <think>\nIn the corrected high-frame-rate clip, the camera zooms in on the boxes being given away. The text on the box is now perfectly clear. **Frame 4 at 21.52s and subsequent frames show the Baltimore Orioles logo and the text** \"BUCK SHOWALTER GARDEN GNOME\" and \"2014 AL MANAGER OF THE YEAR\". The crowd is wearing Orioles gear, and the sign for \"EUTAW STREET\" (frame 4) confirms the location is Oriole Park at Camden Yards. **This confirms all parts of option A**: the team (Orioles), the location (Camden Yards), the item (garden gnome), and the achievement (2014 AL Manager of the Year).\n</think>
<answer>\n\\boxed{A}\n</answer>

Figure 12: Demostration of cold start data.

Based on the video's content, which activity shown in the scenes is **not** one of the restrictions Saudi Arabian women face according to the \"10 Things that Saudi Arabian Women Still Cannot Do\"?
A. Driving a car independently
B. Voting in elections
C. Competing freely in sports
D. Marrying without male guardian consent

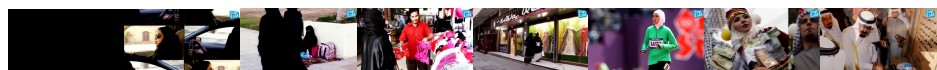

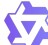 <think>\n1. The video lists \"10 Things That Saudi Arabian Women Still Cannot Do\".\n2. The user wants to identify the activity among the given list that is not one of the restrictions.\n3. I need to confirm what the video explicitly states as a restriction.\n\nLet's examine the provided frames to see which of these activities appear in the video as a restriction:\n- Frame at 7 seconds: Shows a woman driving, which is labeled \"NO. 7\".\n- Frame at 115 seconds: Shows text \"VOTING\".\n- Frame at 144 seconds: Shows women at a market, with text \"MAKE-UP\" on the left.\n- Frame at 194 seconds: Shows women\u8ca0\u3051,f looking at an abaya, with text \"3. WOMEN`S LOYLIY\"\n- Other frames like 150s, 205s, 286s, 317s, 356s, and 362s also introduce other restrictions. However, there is no explicit mention of the specific activity of protected \"white men so they can rightfully protect them\".\n\nThe restriction about men having to be present for certain of women's actions to be protected explicitly mentions men. **It's clear from the provided frames that voting appears explicitly as a restriction** (\"VOTING\" at 115s). This directly answers the question as B. Voting in elections is explicitly mentioned as a restriction, and other options are mentioned previously in the video. I can see the relevant point about voting and confirm it by requesting a higher frame rate clip.\n\nLet's inspect the segment around 100 seconds where \"VOTING\" is first presented. A request for more detailed frames here is appropriate.\n</think>
**<video_zoom>{\"segment\": [101.0, 103.0], \"fps\": 7}</video_zoom>**

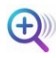 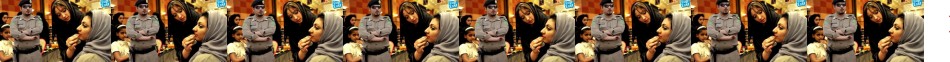

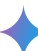 <think>\nThe previous tool call was incorrect because it focused on a completely irrelevant time segment. The call zoomed into 101.0s to 103.0s, which shows a woman getting her makeup done. This segment has no connection to any of the activities listed in the question (driving, voting, sports, marriage). The analysis was therefore based on a flawed premise and a misaligned tool call.\n\nTo answer the question correctly, I must first understand that the video lists all four options\u2014A, B, C, and D\u2014as restrictions. The question asks which one is *not* a restriction, implying that the video itself provides visual evidence to contradict one of its own points.\n\n- **Driving (A)** is listed as a restriction, but the video shows a woman driving, smiling, and making a peace sign (Frame 9, 53.39s).\n- **Voting (B)** is listed as a restriction (Frame 53, 314.48s).\n- **Competing in sports (C)** is listed as a restriction, but the video shows female athletes at the Olympics (Frame 36, 213.61s and Frame 40, 237.34s).\n- **Marrying without consent (D)** is listed as a restriction (Frame 46, 272.94s).\n\nBoth driving and sports show visual contradictions. However, the scene of the woman driving and giving a peace sign is a very direct and celebratory piece of footage that strongly suggests this \"restriction\" is being overcome or is no longer in place. **I will zoom in on this specific moment to confirm the celebratory nature of the action, which would explain why it's the correct answer.** The previous analysis completely missed this.\n\nNow I will zoom in to inspect the motion of 'the woman driving the car' between 52.0s and 54.0s with a higher temporal resolution.\n</think>
**<video_zoom> {\"segment\": [52.0, 54.0], \"fps\": 8} </video_zoom>**

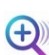 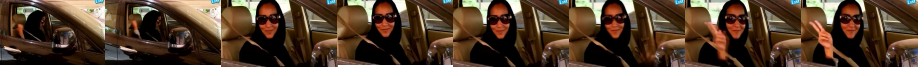 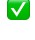

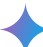 "<think>\nIn the corrected high-frame-rate clip, **I can now clearly see the sequence of events between 52.0s and 54.0s. A woman in a headscarf and sunglasses is sitting in the driver's seat of a car. She turns towards the camera, smiles confidently, and then raises her hand to make a \"peace\" or \"victory\" sign.** The action is deliberate and celebratory. The video presents this scene, along with others of women driving, immediately after listing \"Driving a Car\" as restriction number 7. This visual evidence of a woman driving happily and triumphantly directly contradicts the video's claim that this is something they \"still cannot do.\" While other activities like \"competing in sports\" are also shown, the footage of the woman driving with a peace sign is a more direct and personal refutation of the restriction presented in the video. The other listed restrictions do not have such a clear, contradictory visual representation.\n</think>
<answer>\boxed{A}</answer>"

Figure 13: Example of cold start data.

