# OpenReview forum: "VideoZoomer: Reinforcement-Learned Temporal Focusing for Long Video Reasoning"
_ICLR.cc/2026/Conference — ICLR 2026 Poster_

### Official Review · Reviewer_xgg2 · 2025-10-27

**Soundness:** 3
**Presentation:** 3
**Contribution:** 2
**Rating:** 6
**Confidence:** 4

**Summary:**

This paper proposes an agentic video processing framework: the model is first provided with several frames at a low frame rate and then it can use the temporal zoom tool to obtain high-frame-rate clips at autonomously chosen moments. The framework is worked in a multi-turn manner. The training undergoes a two-phase recipe: a cold-start supervised fine-tuning phase and a reinforcement learning phase. And the authors construct a cold start dataset with 11k samples and diverse reasoning patterns. The resulting model VideoZoomer achieves a remarkable performance improvement over the baseline Qwen2.5-VL 7B.

**Strengths:**

1. The paper constructs a cold start dataset with diverse reasoning patterns, especially the reflection data. Construction reflection data from self-generated failure CoTs is interesting and reasonable, which could be a better way to collect high-quality CoT data.
2. The model is evaluated on a broad range of benchmarks, including long video understanding benchmarks and long video reasoning benchmarks. The robust improvements across multiple benchmarks validate the effectiveness of the proposed methods.
3. The paper also demonstrates that VideoZoomer can further equip a frame selector for frame initialization to improve the performance, showing that the method is orthogonal to other dynamic frame sampling methods.

**Weaknesses:**

1. This work uses LongVideoReason as the training dataset and constructs a cold start training dataset. And the reflection data is constructed from incorrect rollouts. The rollouts are multi-step CoTs, and the errors may occur at any step, CoTs for zoom-in or CoTs for answers.  However, it seems that the LongVideoReason dataset does not have gt timespan annotations. How to determine whether the CoTs for zoom-in are correct or not?
2. The cold start dataset is relatively small.
3. The ablation study on reflection cold start data is not convincing, as more sft data is used.
4. The performance of Qwen2.5-VL 7B baseline on long video understanding benchmarks is relatively low with 128 frames. Can you provide more inference details?
5. The framework is a multi-step reasoning process. A common failure case is that the reasoning can not stop (i.e. the model zooms in on the video endlessly or the model does not provide a valid answer within the budget). How to handle this failure case during training and inference?
6. Compared with other dynamic frame sampling works, this work introduces another sampling variable, 'fps'. And in the provided cases, Figures 4 and 8, fps=8 is extremely high compared to other sampling strategies. Can authors show how the predicted fps affects the performance?
7. The 'fotmat' in Equation (1) is a typo.

**Questions:**

Please see the weaknesses above.

---

> ### Author Response · Authors · 2025-11-21
> **Response to Reviewer xgg2 (Part 1/2)**
>
> We sincerely thank you for your detailed feedback and insightful questions. Below, we address the weaknesses and questions raised.
>
> >**W1: Judging correctness of zoom-in CoTs without GT timespans**
>
> Thank you for raising this crucial point. We use **expert model as judge** instead of gt timespan annotations. For each failed trajectory, we provide the full context (video, question, the agent's flawed reasoning, and the ground-truth answer) to a powerful proprietary model (e.g., GPT-4o). We then prompt it to act as a critic, asking it to:
>
> - Identify the specific step where the reasoning went wrong (e.g., an incorrect zoom, a flawed deduction from a correct clip).
>
> - Generate a corrected reasoning path, including new, more appropriate video_zoom calls if necessary.
>
> To ensure the quality of this generated reflection data, we conducted a **manual inspection** of a subset of the critic's outputs and found that the identified errors and the proposed corrections were logical and led to the correct answer.
>
> >**W2  The cold start dataset is relatively small.**
>
> While our 11,000-trajectory dataset may seem modest in size, its primary purpose is not just to provide examples, but to bootstrap the model's ability to perform complex, multi-step reasoning by exposing it to a wide variety of problem-solving patterns. Our focus was on curating a dataset that is rich in these diverse reasoning pathways.
>
> To investigate the effect of simply increasing the data quantity, we compared the performance of our model trained on our curated 11k dataset against a model trained on a dataset of the same composition but with double the quantity (~20k samples), the results are as follows:
>
> **Table 1: Impact of SFT Data Quantity**
> | Training Dataset | MLVU (dev) | MLVU (test) | LVBench| LongVideoBench| LongVideoReason-eval |
> | --- | --- | --- | --- | --- | --- |
> | Ours (~11k) | 68.8 | 55.8 | 41.5 | 57.7 | 80.3 |
> | Scaled Dataset (~20k) | 69.5 | 54.6 | 41.2 | 55.5 | 78.4 |
>
> As the table demonstrates, the model trained on the larger but less diverse dataset failed to outperform the model trained on our original, more carefully curated set. This indicates that our dataset is effective because it is a high-quality collection designed to efficiently teach a powerful agentic policy, not because of its size.
>
> This finding is philosophically aligned with recent findings in the field. For instance, DeepSeek-R1[1] and LIMO[2] demonstrated that thousands or even just hundreds of high-quality, reasoning-focused SFT samples can be sufficient to unlock strong reasoning capabilities in large models. This "less is more" principle is central to our methodology. Instead of simply scaling up the number of "exemplar" trajectories, which risks overfitting to a dominant, shallow reasoning style, we focused on curating a dataset rich in diverse reasoning patterns.
>
> >**W3 The ablation study on reflection cold start data is not convincing, as more SFT data is used.**
>
> We appreciate you for pointing out this important detail. To properly isolate the impact of reflection data, we conducted a new controlled ablation study, in this new experiment, we ensured the total number of SFT trajectories was identical for all models:
>
> **Table 2: Controlled Ablation Study Results**
> | Model | Total SFT Data | MLVU (dev) | MLVU (test) | LVBench | LongVideoBench | LongVideoReason-eval |
> | --- | --- | --- | --- | --- | --- | --- |
> | Ours | 11k | 68.8 | 55.8 | 41.5 | 57.7 | 80.3 |
> | w/o Reflection | 11k | 67.0 | 53.2 | 40.9 | 54.8 | 75.1 |
> | w/o RL | 11k | 56.4 | 45.6 | 26.0 | 42.0 | 63.3 |
> | w/o cold-start | - | 57.0 | 42.8 | 35.5 | 43.5 | 59.6 |
> | w/o R_tool | 11k | 67.5 | 52.2 | 40.6 | 56.2 | 79.9 |
>
> As the results clearly demonstrate, even when the total amount of SFT data is identical, the model trained with reflection data significantly outperforms the one trained on purely exemplar trajectories. This confirms that the substantial performance gains are not due to the quantity of data but are a direct result of the quality and diversity introduced by the reflection data.
>
> We will revise our paper accordingly to include this more precise and controlled experiment.

---

> ### Author Response · Authors · 2025-11-21
> **Response to Reviewer xgg2 (Part 2/2)**
>
> >**W4: Low baseline performance and inference details**
>
> We appreciate the reviewer's request for clarification. We acknowledge that our reported baseline scores for Qwen-2.5-VL may differ from those in its official technical report[3]. This discrepancy arises because reproducing results from technical reports can be challenging due to ambiguities in specific video inference configurations (e.g., exact frame sampling strategy, image resolution, prompting techniques).
>
> To ensure a **fair and controlled comparison**, we chose to re-evaluate important baselines, including Qwen-2.5-VL, under a single, consistent setting. The configuration used for the base model in our paper is:
>
> - **Frames**: up to 128 frames, uniformly sampled.
> - **Resolution**: Resized every frame to fit 128\*28\*28=100352 pixels.
> - **Prompting**: No CoT or specialized prompting was used, to measure the model's raw capability.
>
> By keeping the evaluation setup identical for both the baseline and our model, we can confidently attribute the performance gains to our proposed VideoZoomer framework. We will make our evaluation code and scripts publicly available to ensure full transparency and reproducibility.
>
> >**W5: How to handle the failure case where the model endlessly zooms in and does not stop?**
>
> This is a common challenge in agentic systems. We handle it through constraints and learning:
>
> - **Training**: During the RL phase, we enforce **a maximum number of tool calls per problem**. If the model exceeds this limit without producing a final answer, the trajectory is considered a failure and receives a significant negative reward. This explicitly teaches the model to provide a definitive answer within the allocated budget. While models in the early SFT stage might exhibit this failure mode, the RL training effectively curtails it.
> - **Inference**: During inference, we apply the same **hard limit** on tool calls. If the limit is reached, we use a **specialized prompt** that instructs the model to stop interacting, summarize all the evidence gathered so far, and provide the best possible final answer.
>
> Empirically, **our final model rarely exceeds the turn limi**t, having learned to converge on an answer with in the limit.
>
> >**W6 fps are high compared to other sampling methods**
>
> We thank the reviewer for this insightful question regarding the `fps` variable. This is indeed a key aspect of our `video_zoom` tool. We would like to clarify about its design and the model's learned behavior.
>
> First, a crucial detail of our framework is that the fps for the zoom-in clip is **not manually specified but is dynamically generated by the model** as part of its tool call. The model's action is to output a complete tool call, allowing it to decide not only where to look but also how closely to look in the temporal dimension.
>
> Second, the design philosophy behind the "zoom" phase is to enable the model to meticulously examine short, critical video segments. During training, we observed that the model effectively learns to use the tool for this purpose. It typically focuses on clips that are only a few seconds long and learns to request a higher fps (e.g., 2, 4, or 8) to capture fine-grained temporal details that would be missed in the initial low-rate overview. An fps of 8, while high for a whole video, is **reasonable and highly effective for a few-second critical video clip.**
>
> We analyzed the distribution of fps values it chose to generate across thousands of tool calls on our validation set:
>
> **Table 3: Distribution of `fps` Values Chosen by the Model**
> | fps Range | (0, 1] | (1, 2] | (2, 4] | (4, 8] | (8, ∞) |
> | --- | --- | --- | --- | --- | --- |
> | Percentage | 24.9% | 66.2% | 8.2% | 0.6% | <0.1% |
>
> As shown by the result, the model does not default to the highest possible `fps`. Instead, its most frequent choice is a moderate `fps` between 1 and 2. This demonstrates that the model has learned an efficient policy, requesting just enough temporal detail to solve the problem without wasting its frame budget.
>
> >**W7：The 'fotmat' in Equation (1) is a typo.**
>
> Thank you for catching this. We have corrected this typo and will perform another round of proofreading to improve the overall quality of the paper.
>
> **References**
>
> [1]Guo, Daya, et al. "Deepseek-r1: Incentivizing reasoning capability in llms via reinforcement learning." arXiv preprint arXiv:2501.12948 (2025).
>
> [2]Ye, Yixin, et al. "LIMO: Less is More for Reasoning." Second Conference on Language Modeling. 2025.
>
> [3]Qwen, A. Yang, et al. "Qwen2. 5 technical report." arXiv preprint (2024).

---

> ### Comment · Reviewer_xgg2 · 2025-11-26
> **Response to the comment**
>
> Thanks authors for providing rebuttals. Most of my concerns are addressed. The newly provide results and analyses should be included in the updated manuscript. I'm happy to maintain the positive rating.
>
> The results of LongVideoBench and LVBench seem to be swapped in Table 1.

---

> > ### Author Response · Authors · 2025-11-26
> >
> > Thank you for your positive feedback, we are glad our rebuttal addressed your concerns.
> > We will incorporate the new results and analyses into the updated manuscript as you suggested. We also appreciate you pointing out the swapped results in Table 1; we have corrected this error.

---

### Official Review · Reviewer_mPz9 · 2025-10-30

**Soundness:** 3
**Presentation:** 3
**Contribution:** 2
**Rating:** 4
**Confidence:** 3

**Summary:**

VideoZoomer is an agentic long-video QA framework that starts with a coarse “glance” and then issues targeted zoom calls over selected time spans to gather fine-grained evidence before answering. It’s trained in two stages—cold-start supervised trajectories (including reflection to correct failures) followed by reinforcement learning with rewards that encourage accurate, well-formatted, and purposeful tool use. Experiments across diverse long-video understanding and reasoning benchmarks show consistently better accuracy–efficiency trade-offs than single-pass baselines, and the approach remains compatible with stronger initial frame selectors.

**Strengths:**

1. This paper introduces a clear agentic framework that couples coarse “glance” perception with targeted temporal zooming, yielding a principled separation between broad coverage and fine-grained evidence acquisition.

2. This paper evaluates across diverse long-video understanding and reasoning benchmarks, with the largest gains on tasks that require precise temporal detail, supporting the method’s intended use case.

**Weaknesses:**

1. The zoom tool is basically one-dimensional: A lot of long-video questions hinge on tiny textual clues (scoreboards, signs, on-screen text), and just “adding frames” won’t reliably capture those.

2. The cold-start data comes from external frontier models (e.g., GPT-style teachers). That brings possible style bias

3. Multi-round zooming can be expensive in practice.

**Questions:**

1. How does accuracy change with 0/1/2/3/4 zoom calls? Is there a point of diminishing returns?

2. When the model zooms the wrong time span, does it recover in later rounds, or does it lock in and fail?

3. How often does the model answer correctly without making any zoom calls, and what types of questions and accuracy are those?

---

> ### Author Response · Authors · 2025-11-21
> **Response to Reviewer mPz9 (Part 1/2)**
>
> We sincerely thank you for your detailed feedback and insightful questions. Below, we address the weaknesses and questions raised.
>
> >**W1: The zoom tool is one-dimensional and may not capture tiny textual clues, as it only adds frames.**
>
> We agree that our current zoom tool primarily addresses the **temporal dimension** of video understanding, which is a principal challenge in many long-video understanding tasks (e.g., identifying the sequence of events, locating specific actions)[1].  Our goal was to first solve this fundamental temporal challenge effectively.
>
> In practice, we found that the standard input resolution (e.g., 336x336) was sufficient for the model to perceive the necessary fine-grained details within the zoomed-in clips. We agree that integrating a spatial zoom capability (e.g., cropping regions of interest) is a promising and complementary future direction. Our agentic framework is designed to be inherently extensible and is orthogonal to such methods, like a spatial zoom tool or as shown in Sec 4.3, a frame selector. However, we believe that mastering the temporal axis is the foundational first step for long video reasoning, which this work successfully demonstrates.
>
> >**W2: Style bias from teacher models**
>
> This is a valid and important point. The potential for "style bias" from proprietary teacher models is a well-known challenge in dataset distillation. We acknowledge this limitation. However, our two-stage training strategy was designed specifically to mitigate this issue:
>
> 1. **Mitigating Bias with On-Policy SFT Data**: The SFT phase is not a simple one-off imitation. We first use the distilled data from the teacher model for a "cold-start," which bootstraps basic tool-use syntax. Crucially, **we then supplement this with on-policy SFT data**, including our "reflection" trajectories generated from the model's own rollouts. By fine-tuning on its own successes and failures, the model begins to learn its own effective patterns, rather than purely mimicking the external teacher's style. This step is critical for diversifying the training data and reducing the initial style bias.
>
> 2. **RL for Policy Optimization**: The subsequent RL phase allows the model to move beyond simple imitation. By interacting with the environment and receiving direct reward signals based on task success, the agent refines its own policy. This process encourages the model to discover strategies that are effective, even if they deviate from the teacher's original style.
>
> In essence, while the initial "style" is influenced by the teacher, the final policy is optimized for performance on the task itself, reducing the model's dependence on the initial distilled trajectories.
>
> As detailed in Appendix C, we trained separate models using datasets distilled from two different powerful teachers: GPT-4o and Gemini 2.5 Pro. **Both models achieved strong and comparable performance** after the full two-stage training process. This demonstrates that our framework can leverage different teaching styles to converge to a highly effective final policy, indicating that ultimate performance is not overly sensitive to the initial "style bias."
>
> >**W3: Multi-round zooming is expensive**
>
> We agree that inference cost is an important practical consideration. However, the cost of our multi-round approach is manageable and often **more efficient** than previous method (i.e., processing a large, fixed number of frames):
>
> 1. **Computational Cost**: The primary computational cost in video MLLMs comes from the vision encoder and the cross-attention layers. Our agent starts with a very small number of frames and only processes additional small clips on demand. As shown in Figure 6, our model consistently uses fewer total frames on average than baselines that process a fixed, large set of 128 or 256 frames, leading to lower overall computational load.
>
> 2. **I/O Latency**: In our parctice, the `video_zoom` action is primarily an **I/O-bound operation** (reading video files from memory or disk) rather than a compute-bound one, which can be largely mitigated with a simple caching mechanism.
>
> Therefore, while our method is multi-round, its adaptive and on-demand nature makes it highly efficient, achieving better accuracy with **less computational overhead** compared to brute-force, single-pass methods.

---

> > ### Author Response · Authors · 2025-11-21
> > **Response to Reviewer mPz9 (Part 2/2)**
> >
> > >**Q1 How does accuracy change with 0/1/2/3/4 zoom calls? Is there a point of diminishing returns?**
> >
> > Thank you for this excellent question about the relationship between the number of zoom calls and accuracy. To investigate this, we conducted a series of experiments where we varied the maximum allowed number of tool calls for our model and evaluated its performance across several key benchmarks. The results are summarized below:
> >
> > **Table 1: Performance by Maximum Allowed Tool Calls**
> > | Max Tool Calls | MLVU (dev) | MLVU (test) | LongVideoBench | LVBench | LongVideoReason-eval |
> > | --- | --- | --- | --- | --- | --- |
> > | 0 (No Tool Use) | 65.0 | 52.6 | 52.7 | 38.7 | 64.8 |
> > | 1 | 66.7 | 53.4 | 55.3 | 39.6 | 79.9 |
> > | 2 | 67.8 | 56.6 | 57.0 | 41.4 | 80.2 |
> > | 3 | 68.4 | 56.0 | 56.2 | 41.8 | 81.1 |
> > | 4 | 69.0 | 55.8 | 57.7 | 41.5 | 80.3 |
> >
> > The most significant performance jump occurs when the model is allowed to use the video_zoom tool at least once. On the LongVideoReason benchmark, for example, accuracy improves from 64.8% (no tool) to 79.9% with just one potential tool call. This confirms that our agentic framework is highly effective at finding and utilizing critical evidence that is missed otherwise.
> >
> > Moving from a maximum of one call to two calls yields another consistent and significant performance boost across most benchmarks. For instance, on MLVU-test, accuracy increases from 53.4% to 56.6%, and on LongVideoBench it rises from 55.3% to 57.0%. This demonstrates the value of multi-turn interaction, allowing the model to refine its search or gather additional evidence when the first attempt is insufficient.
> >
> > After allowing two tool calls, the benefits of increasing the maximum number of calls further become marginal. While there are minor gains on some benchmarks when moving to 3 or 4 calls, the improvements are much smaller and not always consistent. Our final model, with a limit of 4, uses this extra budget sparingly for the hard problems, but the core performance is achieved within the first two interactions.
> >
> > >**Q2 When the model zooms the wrong time span, does it recover in later rounds, or does it lock in and fail?**
> >
> > The model demonstrates a **strong capability for recovery and self-correction**. As shown in the qualitative case studies in our paper (e.g., Figure 9), the model exhibits diverse and sophisticated reasoning patterns. It is capable of retrying with a different time span if its initial zoom does not yield the required information.
> >
> > We attribute this crucial recovery ability to the reflection data included in our SFT phase. We performed an ablation study where we trained a model without this reflection data. That model was far more likely to "lock in" on its initial, incorrect zoom and ultimately fail the task. The reflection trajectories explicitly teach the model how to recognize a failed attempt and formulate a new plan, making it more robust.
> >
> > >**Q3 How often does the model answer correctly without making any zoom calls, and what types of questions and accuracy are those?**
> >
> > Thank you for this excellent question about the model's behavior in zero-call scenarios. Our analysis shows that in over 90% of cases across our evaluated benchmarks, the final model opts to make at least one `video_zoom` call before providing a final answer.  This behavior demonstrates that the model learned a highly cautious and evidence-driven policy: for complex long-video questions, the initial low-frame-rate "glance" is generally insufficient for a reliable answer. Instead of making a premature guess, it has learned to seek higher-quality visual evidence to verify its answer.
> >
> > In the rare instances where the model answers without a tool call, the questions are typically about the overall topic or main subject of the video.  For these global-level questions, the initial sparse frames can be sufficient and the model achieves relatively high accuracy on these questions.
> >
> > **References**
> >
> > [1]Zou, Heqing, et al. "From seconds to hours: Reviewing multimodal large language models on comprehensive long video understanding." arXiv preprint arXiv:2409.18938 (2024).

---

> ### Author Response · Authors · 2025-11-27
>
> Dear Reviewer mPz9:
>
> Thank you again for your thoughtful feedback and for taking the time to review our work.
>
> We hope that our responses have addressed your concerns. As the rebuttal deadline is approaching, we are writing to kindly check whether our replies have resolved your questions or if any points would benefit from further discussion.
>
> Please feel free to let us know if any further questions arise.

---

### Official Review · Reviewer_JYRh · 2025-11-01

**Soundness:** 2
**Presentation:** 2
**Contribution:** 1
**Rating:** 4
**Confidence:** 5

**Summary:**

The paper introduces VideoZoomer, an agentic model enabling MLLMs to dynamically adjust their visual focus during reasoning. Beginning with a low-frame-rate overview, the model uses a temporal zoom tool to capture high-frame-rate clips at key moments, progressively gathering fine-grained evidence. The training process involves supervised fine-tuning on curated datasets, followed by optimization with DAPO. Experimental results demonstrate that the 7B model exhibits diverse and sophisticated reasoning capabilities, achieving strong performance in long video understanding tasks.

**Strengths:**

- The paper contributes a training dataset comprising 11,000 trajectories, which is used to enhance the tool-calling capabilities of models.

- The case visualizations presented in the paper are good

**Weaknesses:**

- The technical contributions of the paper are limited, as its main novelty lies in providing a curated training dataset to enhance the tool-calling capabilities of models.

- The method adds a bonus to tool-call rewards when the final answer is correct, which increases the unnecessary frequency of tool use. For example, the model may continue calling tools unnecessarily, retrieving irrelevant clips even after it already has the correct answer, ultimately inflating the reward.

- While the paper mentions using 11,000 trajectories for the cold-start phase of training, it does not specify the data used during the RL phase, leaving gaps in transparency and reproducibility.

**Questions:**

- Will further scaling up the training dataset continue to improve the model’s performance?

- The paper mentions starting with 64 frames as input, gradually increasing by 16 frames each time. However, in Figure 1 (the right part), the number of input frames for VideoZoomer seems inconsistent with this description.

- Will the code and the training dataset used in the paper be open-sourced?

---

> ### Author Response · Authors · 2025-11-21
> **Response to Reviewer JYRh (Part 1/2)**
>
> We sincerely thank you for your detailed feedback and insightful questions. Below, we address the weaknesses and questions raised.
>
> >**W1:Limited technical contributions**
>
> We would like to clarify the technical contributions of our work. While the curated dataset is a valuable asset, our primary contributions lie in the agentic framework and the associated training methodology.
>
> 1. **Agentic Reasoning Framework**: VideoZoomer is a framework that empowers MLLMs with the ability to dynamically control their visual focus during reasoning.  Instead of passively receiving a fixed set of frames, our model **actively decides where to look and how closely to look in a multi-step process**. This moves beyond static, single-pass methods (like uniform or pre-selected sampling) which cannot recover from initial error. Our agentic approach allows the model to correct its own focus and progressively gather evidence, achieving a significant advancement in reasoning capability.
>
> 2. **Specialized Training Strategy**: Our two-stage training strategy is crucial for enabling this agentic behavior. The key innovations are:
>
> - **Bootstrapping with "Reflection"**: We train the model not just on successful examples, but also on its own failures. By learning from "reflection" trajectories, the model learns how to recover from mistakes—a crucial skill for multi-step reasoning that is difficult to teach otherwise.
>
> - **RL-based Policy Refinement**: The subsequent Reinforcement Learning (RL) phase with a carefully designed reward function further refines the agent's policy, optimizing for accuracy and efficiency, moving beyond simple imitation learning.
>
> We believe this combination of agentic reasoning framework and a specialized training pipeline represents a substantial technical contribution to the field of long video understanding.
>
> >**W2: The reward bonus for correct final answers may encourage unnecessary tool use**
>
> We sincerely thank the reviewer for this insightful and valid point. The potential for encouraging redundant actions is a critical challenge in designing RL agents.
>
> First, we found that the model learns an efficient policy in practice. Across our evaluation benchmarks, **over 80% of questions are answered with just one or two tool calls.** The model resorts to more calls only for a small fraction of complex problems, demonstrating it has learned to stop once sufficient evidence is gathered.
>
> During our experiments, we found that including a tool-use bonus was **essential for the model to learn how to use the tool effectively.** As shown in our ablation study (Table 3 and Figure 5), the model trained without the `R_tool` bonus (`w/o R_tool`) suffered from **"policy collapse**." It learned to avoid using the tool altogether, leading to a drastic drop in performance as it failed to explore and discover the tool's utility. At the same time, we were mindful of the potential for reward hacking and implemented a key safeguard: the tool bonus is **conditional**. It is only awarded if the final answer of the entire trajectory is correct. This means a trajectory with superfluous tool calls that ultimately leads to a wrong answer will receive no bonus, which disincentivizes random or unhelpful tool use.
>
> We acknowledge that teaching the model to perfectly and adaptively decide when to stop using tools—especially when it might already have sufficient information—is a highly complex challenge. While our current framework learns an effective policy, optimizing this "stopping criteria" was not the primary focus of this paper. We believe that developing more sophisticated reward schemes or training methods to better handle this adaptive stopping problem is a valuable and important direction for future research.
>
> >**W3: The data used during the RL phase is not specified.**
>
> We apologize for this lack of clarity. For the reinforcement learning phase, we used the **same dataset(LongVideoReason[1])** as in the Supervised Fine-Tuning (SFT) phase. We will explicitly state this in the next version and release all data, code and models publicly to improve clarity and reproducibility.

---

> > ### Author Response · Authors · 2025-11-21
> > **Response to Reviewer JYRh (Part 2/2)**
> >
> > >**Q1 Will further scaling up the training dataset continue to improve the model’s performance?**
> >
> > We thank you for raising these important questions about the size and scalability of our training dataset. Our methodology is philosophically aligned with recent findings suggesting that for complex reasoning tasks, **data quality can be more critical than sheer quantity.** For instance, LIMO[2] demonstrated that thousands or even just hundreds of high-quality, reasoning-focused SFT samples can be sufficient to unlock strong reasoning capabilities in large models. This "less is more" principle is central to our methodology. Instead of simply scaling up the number of "exemplar" trajectories, which risks overfitting to a dominant, shallow reasoning style, we focused on curating a dataset rich in diverse reasoning patterns.
> >
> > To directly investigate the impact of both quantity and quality (diversity), we compared our final model against two variants: one trained on a dataset of the same composition but with double the quantity, and another trained on the same quantity but with less reasoning diversity (i.e., without "reflection" data):
> >
> > **Table 1: Impact of SFT Data Quantity and Diversity**
> > | Training Dataset | Data Size  | MLVU (dev) | MLVU (test) | LVBench | LongVideoBench | LongVideoReason-eval |
> > | --- | --- | --- | --- | --- | --- | --- |
> > | Scaled Dataset | ~20k  | 69.5 | 54.6 | 41.2 | 55.5 | 78.4 |
> > | w/o Reflection | ~11k | 67.0 | 53.2 | 40.9 | 54.8 | 75.1 |
> > | Ours (Final Model) | ~11k  | 68.8 | 55.8 | 41.5 | 57.7 | 80.3 |
> >
> > As the table demonstrates, the model trained on the larger dataset failed to outperform the model trained on our original, more carefully curated set. With the exact same data volume (~11k), the model trained on a more diverse dataset including "reflection" trajectories significantly outperforms the model trained only on standard exemplars trajectories.
> >
> > Therefore, while our 11k-trajectory dataset may seem modest in size, it is a high-quality, targeted collection designed to efficiently bootstrap a powerful agentic policy. Answering the question of whether further scaling would help: yes, but only if the new data introduces even more diverse and complex reasoning strategies.
> >
> > >**Q2: Inconsistency between the described frame numbers (start with 64, add 16) and the numbers in Figure 1.**
> >
> > Thank you for pointing out this potential confusion. The frame counts mentioned in the text (e.g., an initial 64 frames and 16 frames per zoom, as detailed in Table 1) represent the **upper bound** of the frame budget allocated to our model. This was designed to ensure a fair comparison with baselines, guaranteeing that our model does not exceed the 128-frame budget used by other methods.
> >
> > In practice,  our model operates dynamically and often solves a problem without exhausting its full budget. For instance:
> >
> > - A short video may not have enough frames to sample the full 64-frame overview.
> >
> > - The model might find the correct answer after just one `zoom` action and terminate its reasoning process early.
> >
> > The numbers shown in Figure 1 and Figure 6 represent the **average number of frames actually used** by our model across the benchmark. This average number is naturally lower than the maximum theoretical budget. We will revise the figure captions and relevant text to make this distinction between "maximum budget" and "actual usage" clearer.
> >
> >
> > >**Q3: Will the code and the training dataset be open-sourced?**
> >
> > Yes. We are committed to contributing to the research community. The code, dataset and models used in this paper will be made publicly available.
> >
> > **References**
> >
> > [1]Chen, Yukang, et al. "Scaling RL to Long Videos." The Thirty-ninth Annual Conference on Neural Information Processing Systems. 2025.
> >
> > [2]Ye, Yixin, et al. "LIMO: Less is More for Reasoning." Second Conference on Language Modeling. 2025.

---

> ### Author Response · Authors · 2025-11-27
>
> Dear Reviewer JYRh:
>
> Thank you again for your thoughtful feedback and for taking the time to review our work.
>
> We hope that our responses have addressed your concerns. As the rebuttal deadline is approaching, we are writing to kindly check whether our replies have resolved your questions or if any points would benefit from further discussion.
>
> Please feel free to let us know if any further questions arise.

---

> > ### Comment · Reviewer_JYRh · 2025-11-28
> > **Response to the comment**
> >
> > Thank you for providing the rebuttal. Some of my concerns have been addressed. However, my major concerns still remain.
> >
> > For novelty, I believe the proposed `<video_zoom>` tool simply adds another tool to LLM's tool-calling mechanism, which does not constitute a solid contribution. However, I appreciate the authors' consideration of the reflection mechanism, which I agree is important.
> >
> > 1. Regarding tool rewards: In existing LLM tool-use designs, such as web search, there is no additional tool reward granted for trajectories based on whether they employ tool calls. The "policy collapse" issue mentioned by the authors stems from training data where not all questions benefit from `<video_zoom>` to achieve a correct answer. This predisposes the model away from using the tool. Moreover, errors in zoom-in intervals might also lead the model to generate incorrect answers, lowering the rewards for the trajectories with `<video_zoom>`.
> >
> > Furthermore, for the same queries where both `<video_zoom>` and non-`<video_zoom>` trajectories yield correct answers, it is generally preferable to save token computation by encouraging non-`<video_zoom>` trajectories. The tool reward designed in this paper simply introduces a bonus for trajectories with correct final answer, rather than whether the zoom-in intervals are correct for these trajectories. This is analogous to a "length reward or penalty," which only incentivizes adding intermediate tokens when producing correct answers. This design could inadvertently encourage models to use unnecessary tool calls. This is my core concern.
> >
> > 2. Regarding the training data: For the cold-start and RL phases, the authors mention using the LongVideoReason dataset. Were the same questions used in both phases, or was the dataset further segmented? Regarding the point on scaling up, I agree with the authors that the quality and diversity of patterns are more important than sheer quantity. However, when following the same data generation pipeline, the quality should be relatively consistent. The observation that performance decreases when scaling up to 20k instances, while 11k performs better, suggests that determining an optimal quantity for cold-start training data is highly uncertain. How did the authors determine that 11k is the "just right" quantity for cold-start training?
> >
> > 3. Regarding zoom-in segments: From the examples provided by the authors, I noticed that most zoom-in segments are 1 or 2 seconds. How does the model handle situations where the relevant segment extends beyond 2 seconds or involves multiple video segments?
> >
> > In summary, while the rebuttal has addressed some minor concerns, I believe the core issues of contribution and tool reward design remain concerns.

---

> > > ### Author Response · Authors · 2025-12-01
> > >
> > > Dear Reviewer JYRh,
> > >
> > > Thank you for your detailed and thoughtful follow-up. We would like to further address your concerns point by point.
> > >
> > > ### 1. Regarding the Tool Reward Design
> > >
> > > We thank you for this critical point. We argue that our tool's novelty and complexity distinguish it from common tools like web search or code, which are already present in pre-training data. Our early RL experiments showed that even when trained exclusively on problems impossible to solve without the tool, the model still defaulted to a locally optimal strategy and failed to learn a reliable tool-use policy without an explicit reward. The `R_tool` bonus acts as a necessary exploration incentive to overcome this deep-seated habit.
> > >
> > > Regarding rewarding only correct intermediate steps, we believe that allowing for some trial-and-error within a trajectory is beneficial, as it encourages the development of more complex, multi-step reasoning patterns. Therefore, we reward the entire trajectory based on the final correct outcome. To prevent excessive exploration, we enforce a strict limit on the maximum number of tool calls, striking a balance between encouraging sophisticated reasoning and ensuring efficiency. As demonstrated by our experimental results, this approach leads to a more efficient and effective policy.
> > >
> > > ### 2. Regarding the Training Data
> > >
> > > **On SFT and RL Data Selection:**
> > > We first assessed the difficulty of each question by measuring the correctness rate of our base model over 8 rollouts. For the SFT phase, we strategically used the easier questions to generate exemplar trajectories and the harder questions to generate reflection trajectories. Subsequently, the RL phase utilized the entire training dataset. We did not need to pre-filter this data because we employed the dynamic sampling mechanism from DAPO [1], which automatically adapts the training curriculum by filtering out samples that are too easy or too hard for the current policy, thereby ensuring maximum training efficiency. We will add this clarification to the paper.
> > >
> > > **On Scaling the Dataset:**
> > > Our primary goal was not to argue that 11k is the best size for a cold start, but to demonstrate that it was sufficient in our specific context. The performance drop at 20k suggests that merely adding more data of the same type does not guarantee improvement. It may introduce redundancy that leads to minor overfitting on certain patterns, slightly harming generalization. This suggests that for complex agentic tasks, future efforts should be prioritized not on simply scaling the dataset size, but on more sophisticated strategies like enhancing data diversity and improving the quality of the reflection data.
> > >
> > > ### 3. Regarding Zoom-in Segments
> > >
> > > You are correct that our examples often feature short zoom-in segments. This is a direct result of our model learning an efficient request strategy from the training data, which is rich in fast-paced scenes where critical information is concentrated in brief moments. The model learns to request short, high-frame-rate clips because they are often sufficient to answer the query, demonstrating its ability to retrieve information effectively.
> > >
> > > For longer-duration events, the necessary context is typically captured in the initial video overview. However, our framework is fully capable of handling exceptions; the model can and does request longer segments when needed. To provide concrete evidence, we will add case studies to the appendix showing the model successfully requesting longer video segments.
> > >
> > > ---
> > > [1] Yu, Qiying, et al. "Dapo: An open-source llm reinforcement learning system at scale." arXiv preprint arXiv:2503.14476 (2025).

---

### Official Review · Reviewer_21rm · 2025-11-01

**Soundness:** 4
**Presentation:** 3
**Contribution:** 3
**Rating:** 8
**Confidence:** 3

**Summary:**

This paper investigates the video LLM with self-bootstrapped clip selection capability by SFT and RL to reduce the frame sampled and inference cost. This paper presented VideoZoomer, a agentic framework integrating <video_zoom> tool call and <think> thinking abilities, to select video clips for its processing. The proposed VideoZoomer method can reach outperforming accuracies compared to previous methods while using less input frames. The author also performed detailed ablation studies on cold-start, reflection, RL, and score design.

**Strengths:**

1) This paper is one of the first cohort to explore the frame selection method by agentic RL in video LLM domain, delivering substantial novelty. Also, the agentic RL framework is not simply adopted from the image/language domain to the video domain. Instead, it incorporates tool calling and methods designed specifically for videos (e.g., temporal zoom-in, on-policy reflection).

2) The resulted agentic model by GRPO training, outperforms the baseline model by a large margin, especially on the long video understanding task. The author also detail key components in proposed RL framework, showing that the cold-start and reflection finetuning is essential for good RL models in reasoning video frame selection model, which is of great valuable insights.

**Weaknesses:**

Overall this paper is of great technical value and soundness, but there are some minor concerns listed below:

1) There are several minor typos across texts, including line 151: "stragety", line 277: "fotmat" and more.  The author should perform grammar and word check throughout the paper;

2) Are other capabilities of video LLMs (Qwen 2.5-VL) well maintained? Like short video captioning?

3) Qwen 2.5-VL is known to lack of native <think></think> reasoning capabilities. The authors performed off-policy warm-start and on-policy reflection SFT to enable the reasoning capability but is it robust when the input data is out of distribution of SFT training data, like very simple CLEVRER data? Just curious and I want to hear the authors' insights into it.

**Questions:**

See weaknesses.

---

> ### Author Response · Authors · 2025-11-21
> **Response to Reviewer 21rm**
>
> We sincerely thank you for your detailed feedback and insightful questions. Below, we address the weaknesses and questions raised.
>
> >**W1: Minor typos across the text.**
>
> Thank you for pointing these out. We sincerely apologize for these oversights. We have performed a thorough grammar and spell-check of the entire manuscript and will correct the typos and other errors in next version.
>
> >**W2: Are other capabilities of video LLMs (Qwen 2.5-VL) well maintained? Like short video captioning?**
>
> This is an excellent question regarding the generalizability of our fine-tuned model. While tasks on short videos are not the primary focus of our work, we conducted experiments on short video captioning benchmarks and found that `VideoZoomer` not only maintains but shows non-trivial improvements over the baseline's performance.
>
> **Table 1: Short Video Captioning Benchmark Results**
> | Model | TemporalBench (Short Caption Score) | TempCompass (Captioning Acc) | VDC (Short Acc / Score) |
> | --- | --- | --- | --- |
> | Qwen-2.5VL (Baseline) | 40.9 | 52.1 | 37.8 / 1.98 |
> | VideoZoomer (Ours) | 56.4 | 65.3 | 49.2 / 2.51 |
>
> We attribute this improvement to the high-quality SFT data used in our training pipeline, which appears to enhance the model's overall video-language alignment and descriptive abilities.
>
> Our main goal was to enhance long-video capabilities without harming others, and these results confirm we have achieved this while also **boosting the model's general descriptive capabilities.**
>
> >**W3: Robustness on out-of-distribution (OOD) data like CLEVRER**
>
> We agree that testing on out-of-distribution data is crucial for understanding the true capabilities and limitations of our model.
>
> To investigate the model's robustness on a task that falls outside this primary design scope, we evaluated our model on a subset of the CLEVRER dataset[4]. This benchmark tests causal and logical reasoning on synthetic short videos, a domain that is significantly different from the real-world, long-form videos used in our training.
>
> **Table 2: Performance on CLEVRER**
> | Model | CLEVRER Accuracy |
> | --- | --- |
> | Qwen-2.5-VL (Base) | 67.3% |
> | VideoZoomer (Ours) | 68.0% |
>
> This result shows that:
> - **Our training process does not degrade foundational abilities.** The comparable performance to the baseline demonstrates that our two-stage training successfully r**etains the model's core
> competencies** on tasks that do not require our agentic framework.
> - **The minimal gain is expected as the task is out-of-scope**. Our VideoZoomer framework, with its "glance-and-zoom" mechanism, is specifically designed to address the core challenges of long video understanding, where efficiently navigating extensive temporal context is paramount. The skills for that task do not directly transfer to the abstract, logical reasoning tested by CLEVRER.
>
> **References**
>
> [1]Cai, Mu, et al. "Temporalbench: Benchmarking fine-grained temporal understanding for multimodal video models." arXiv preprint arXiv:2410.10818 (2024).
>
> [2]Liu, Yuanxin, et al. "TempCompass: Do Video LLMs Really Understand Videos?." Findings of the Association for Computational Linguistics ACL 2024. 2024.
>
> [3]Chai, Wenhao, et al. "AuroraCap: Efficient, Performant Video Detailed Captioning and a New Benchmark." ICLR 2025. 2025.
>
> [4]Yi, Kexin, et al. "Clevrer: Collision events for video representation and reasoning." arXiv preprint arXiv:1910.01442 (2019).

---

> ### Author Response · Authors · 2025-11-27
>
> Dear Reviewer 21rm:
>
> Thank you again for your thoughtful feedback and for taking the time to review our work.
>
> We hope that our responses have addressed your concerns. As the rebuttal deadline is approaching, we are writing to kindly check whether our replies have resolved your questions or if any points would benefit from further discussion.
>
> Please feel free to let us know if any further questions arise.

---

### Author Response · Authors · 2025-12-02
**To ACs and Reviewers**

Dear ACs and Reviewers,

We sincerely thank all reviewers for their time and for providing detailed, constructive feedback that has been invaluable in strengthening our work. We have carefully considered all comments and have revised our manuscript accordingly. We have also provided detailed point-by-point responses to each reviewer. We would like to highlight the following points.

1.  **Strengths Highlighted by Reviewers**
- **Novelty and Significance**: The paper is one of the first to explore agentic RL for frame selection in video LLMs, with a framework specifically designed for video, not just adapted from other domains (Reviewer 21rm). The agentic framework coupling a "glance" with targeted "zooms" was seen as clear and principled (Reviewer mPz9).
- **Strong Empirical Results**: The reviewers acknowledged that our model, VideoZoomer, significantly outperforms baselines, especially on long video tasks (Reviewer 21rm), and that the robust improvements across a broad range of benchmarks validate the method's effectiveness (Reviewer xgg2).
- **Valuable Methodological Insights**: The detailed ablation studies were appreciated, showing that the cold-start and reflection mechanisms are essential for training a powerful agent (Reviewers 21rm, xgg2). The construction of reflection data from self-generated failures was noted as an interesting and reasonable approach (Reviewer xgg2).

2. **Revisions and Additional Experiments**

To address the reviewers' insightful questions, we have performed several new experiments and incorporated the following key additions and clarifications into our revised manuscript.
- **Robustness on Out-of-Distribution Task (Reviewer 21rm)**: We conducted a new experiment on the CLEVRER dataset. The results show that our training does not harm the base model's foundational capabilities on unrelated tasks, demonstrating that we have added a new skill without causing catastrophic forgetting.
- **Generalization to Short-Video Tasks (Reviewer 21rm)**: We provided new results on short-video captioning benchmarks, showing that VideoZoomer not only maintains but significantly improves upon the baseline's performance, indicating an enhancement of the model's general descriptive capabilities.
- **Controlled Ablation Study on Reflection Data (Reviewer xgg2)**: To provide a more convincing ablation, we conducted a new controlled experiment ensuring the total SFT data volume (11k) was identical for the model trained with and without reflection data. The results decisively show that the performance gain comes from the quality and diversity of reflection data, not data quantity.
- **Analysis of fps (Reviewer xgg2)**: We provided a detailed analysis of the fps values dynamically generated by the model, showing that it learns an efficient policy, defaulting to a moderate fps (1-2) in 66% of cases and only escalating to higher fps when necessary. This demonstrates the model's adaptive, rather than brute-force, sampling strategy.
- **Analysis of Dataset Scaling (Reviewer JYRh, xgg2)**: We conducted new experiment to demonstrate that for our complex agentic learning task, data diversity and quality are more critical than sheer quantity.
- **Analysis of Performance vs. Number of Tool Calls (Reviewer mPz9)**: We conducted a new experiment analyzing the relationship between the maximum allowed tool calls and model accuracy. The results show a significant performance leap with one tool call, another consistent boost with two calls. This demonstrates the value of tool calling and multi-turn interaction.

We have added these new experiment results into our revised manuscript,  all revisions and newly added materials are clearly marked in blue.

We believe these additions and clarifications have substantially strengthened the paper. We respectfully invite the reviewers and AC to consider our responses and revisions, and we remain open to any further discussion.

Best Regards,

Authors of Submission #10800

---

### Meta-Review · Area_Chair_mhQi · 2026-01-05

**Summary:**

The reviews present a mixed scores  of the submission （8/4/4/6）.

Reviewer 21rm is strongly positive (score: 8), finding the approach technically sound and novel in its agentic RL formulation for long-video understanding. Reviewer xgg2 is moderately positive (score 6), acknowledging the effectiveness of the framework while raising concerns about data construction, training details, and failure handling. In contrast, Reviewers JYRh and mPz9 are more critical (both score 4), questioning whether the primary contribution extends beyond a curated training dataset and an engineered combination of existing agentic tool-calling components, and expressing concerns about reward design, reproducibility, analysis of tool usage behavior, and practical efficiency. Across reviews, there is general agreement on empirical effectiveness, with disagreements mainly centered on the level of methodological novelty, transparency of the RL setup, and the depth of analysis on zoom behavior and cost trade-offs, alongside minor presentation issues.

**Reviewer Concerns:**

Reviewer 21rm: Minor concerns about presentation quality and whether the proposed training affects generalization or robustness beyond long-video reasoning tasks.

Reviewer JYRh: Questions the level of methodological novelty, viewing the work as an engineering integration of existing agentic and RL components; raises concerns about reward design, reproducibility.

Reviewer mPz9: Focuses on the practical limits of the approach, including whether temporal-only zooming is sufficient, the real-world cost of multi-round zooming, and the lack of detailed analysis of tool usage behavior.

Reviewer xgg2: Generally supportive but concerned about training signal quality and clarity, particularly the correctness of reflection supervision, handling of failure cases, termination criteria for multi-round zooming, and insufficient detail in training and inference settings.

**Reviewer Scores:**

After the rebuttal, Reviewer xgg2 indicated that their concerns were addressed, while Reviewer JYRh maintained that the core issues regarding methodological novelty and tool reward design remain unresolved. Other reviewers did not provide further feedback.

Considering the strong empirical results, I recommend acceptance. The newly provided results and analyses in the rebuttal should be incorporated into the revised manuscript to improve clarity.

---

### Decision · Program_Chairs · 2026-01-26

Accept (Poster)